# Local-Global Coupling Spiking Graph Transformer for Brain Disorders Diagnosis from Two Perspectives

**Geng Zhang**[1,2,3], **Jiangrong Shen**[1,2,4]*, **Kaizhong Zheng**[1,2,3],
**Liangjun Chen**[1,2,3], **Badong Chen**[1,2,3]*
[1]National Key Laboratory of Human-Machine Hybrid Augmented Intelligence
[2]National Engineering Research Center for Visual Information and Applications
[3]Institute of Artificial Intelligence and Robotics, Xi'an Jiaotong University
[4]Faculty of Electronic and Information Engineering, Xi'an Jiaotong University

## Abstract

Brain disorders have been consistently associated with abnormalities in specific brain regions or neural circuits. Identifying key brain regional activities and functional connectivity patterns is essential for discovering more precise neurobiological biomarkers. However, previous studies have primarily emphasized alterations in functional connectivity while overlooking abnormal neuronal population activity within brain regions. To bridge this gap, we propose a novel Local-Global Coupling Spiking Graph Transformer (LGC-SGT) that jointly models both inter-regional connectivity differences and deviations in neuronal population firing rates within brain regions, enabling a dual-perspective neuropathological analysis. The global pathway leverages spike-based computation in LGC-SGT to model biologically plausible aberrant neural firing dynamics, while the local pathway adaptively captures abnormal graph-based representations of brain connectivity learned by local plasticity in the liquid state machine module. Furthermore, we design a shortcut-enhanced output strategy in LGC-SGT with the hybrid loss function to suppress outlier interference caused by inter-individual and inter-center variability, enabling a more robust decision boundary. Extensive experiments on three brain disorder datasets demonstrate that our model consistently outperforms state-of-the-art graph methods in brain disorder diagnosis. Moreover, it facilitates the extraction of interpretable neurobiological biomarkers by jointly analyzing regional neural activity and functional connectivity, offering a more comprehensive framework for brain disorder understanding and diagnosis.

## 1 Introduction

Brain disorders are complex, multi-level conditions characterized by alterations at both the neuronal and network levels [1]. Evidence suggests that excessive neuronal activity in the hippocampus and amygdala, along with reorganization of functional connections within the prefrontal cortex and the default mode network, likely contributes to cognitive impairments [2]. Connectivity patterns within these regions are particularly crucial, as they mediate the integration of neural circuits that underpin cognitive function [3]. Moreover, stress-related disorders, such as major depressive disorder (MDD) and schizophrenia, have been shown to be associated with neural activity in the amygdala [4]. Thus, abnormal activity in specific neuronal populations can disrupt neural oscillations, also leading to cognitive dysfunction [5]. Spiking neural networks (SNNs), which biologically model neuronal firing behaviors in a biologically plausible manner, offer a unique opportunity to integrate

---

*Corresponding author: jrshen@zju.edu.cn, chenbd@mail.xjtu.edu.cn

39th Conference on Neural Information Processing Systems (NeurIPS 2025).

connectivity analysis with population-level neural dynamics. Therefore, this study aims to distinguish patients from healthy individuals using SNNs, considering both neuronal activity and connectivity reorganization at the network level.

Recent advances in SNNs have demonstrated their capability to characterize static functional connectivity patterns[6, 7]. Building on this foundation, recent studies have extended SNN frameworks to model dynamic functional and structural connectivity [8, 9]. However, their analytical paradigms are limited to functional connectivity changes between brain regions, overlooking the representation of neuronal dynamics within brain regions. This limits their completeness in modeling brains. For example, Alzheimer's pathological changes often begin with micro-disruptions in synaptic plasticity within the hippocampus, while abnormalities in macro-level functional connectivity may not become apparent for several years [10]. Therefore, by modeling the dynamic characteristics of neuronal clusters within brain regions alongside functional connectivity differences, we can capture brain changes from both microscopic and macroscopic perspectives, providing more accurate biomarkers. However, existing SNN frameworks seldom integrate these two perspectives, *i.e.* network topology and neuronal population responses, within a single computational model, limiting their ability to disentangle multi-scale biomarkers.

To bridge this gap, we propose a Local-Global Coupling Spiking Graph Transformer (LGC-SGT) that jointly models inter-regional connectivity discrepancies and neuronal population firing rate deviations across brain areas, enabling a dual-perspective analysis of neuropathology. Two critical perspectives are provided by our model: 1) disorder-sensitive alterations in inter-regional connectivity strength and topology, and 2) abnormal firing rate responses of neuronal populations within specific brain regions. Therefore, our framework dynamically links macroscale connectivity patterns with microscale-inspired spiking population dynamics. Our spike-based transformer integrates the local-global coupling module, the spiking transformer feature extraction block, and the shortcut-enhanced output strategy with a hybrid loss function. The local-global coupling module captures the spatiotemporal interactions of both aberrant firing rate and abnormal brain connectivity. The local liquid state machine (LSM) pathway guided by functional magnetic resonance imaging (fMRI) initializes region-specific neuronal populations with random connectivity, then updates synaptic weights via spike-timing-dependent plasticity (STDP) to emulate local plasticity, constructing the functional connection pattern. The global adaptive spiking graph convolution pathway refines inter-regional connectivity weights via Backpropagation and integrates global information to capture the neural firing dynamics. Besides, the shortcut-enhanced output module with the hybrid loss function is designed to obtain a more robust decision boundary by suppressing the interference caused by the outliers since the differences between inter-individual and inter-center.

The contributions of our paper could be summarized as follows:

- We propose the local-global coupling spiking graph transformer framework for brain disorder diagnosis. The spike-based transformer integrates local and global pathways, attending to spatiotemporal interactions of both aberrant firing rate dynamics and abnormal brain connectivity.

- The shortcut-enhanced output strategy with the hybrid loss function is designed to obtain a more robust decision boundary by suppressing the interference caused by the outliers since the variety of inter-individual and inter-center.

- The experiments on three brain disorder datasets show that the proposed LGC-SGT model outperforms existing methods and provides the dual-perspective biomarker discovery of interregional connectivity discrepancies and neuronal population firing rate deviations.

## 2 Related Work

### 2.1 Function-based Brain Networks Analysis

With the rapid advancement of neuroimaging technologies, researchers have progressively established a systematic neuroimaging-based paradigm for investigating brain networks [11], with fMRI emerging as a crucial data foundation for constructing brain networks through its unique integration of whole-brain coverage and high spatial resolution [12]. The complexity of modeling the brain's topological relationships has sparked researchers' interest, leading to the development of methods using Graph Neural Networks (GNN) for brain network analysis [13]. Additionally, approaches such as adaptive

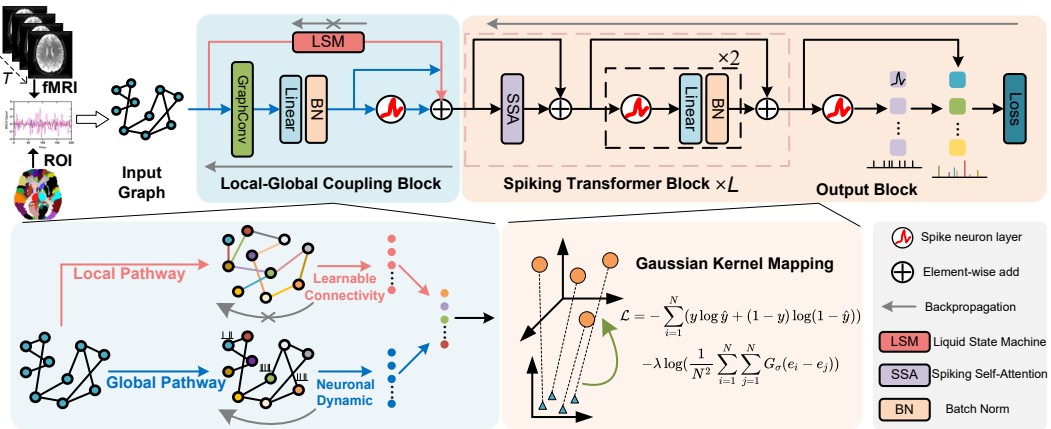

Figure 1: The overall framework of the proposed LGC-SGT model.

brain region discrimination [14], information-theory-driven connectivity selection [15], and long-range dependency modeling [16] have been proposed. However, these methods typically focus on modeling functional connectivity relationships and often overlook the representation of neuronal population dynamics within the brain.

## 2.2 SNN-based Brain Networks Analysis

Recent years have seen growing applications of SNNs in diverse fields [17–19], owing to their biological plausibility and low-power characteristics [20]. In brain cognitive modeling, SNNs enable biologically-based brain network modeling by simulating the dynamic processes of biological neurons. Three-dimensional spiking neural network architectures facilitate the analysis of spatiotemporal neural data through biomimetic topological design [21]. By spatio-temporal associative memories in SNNs, Kasabov et al. [22] achieved brain disorder classification based on fMRI data. The biologically-inspired characteristics of SNNs reveal how mindfulness training reorganizes brain networks [23] and allow for data analysis of the coupling between brain structure and functional connectivity [8, 9]. Additionally, SNNs constrained by fMRI topology uncover auditory coding mechanisms from the perspective of information transfer efficiency [24]. Anatomically-guided spiking networks leverage brain anatomical features to guide SNNs in autonomously learning brain network characteristics [25]. Despite these advancements, current SNN frameworks still have limitations in brain network modeling, particularly in their lack of collaborative modeling that integrates inter-region network topology with the responses of neuronal populations within brain regions.

# 3 Method

## 3.1 Problem Definition

In brain network analysis, the core objective of our model is to predict the presence of a brain disorder or to discover new biomarkers. Here, the network is constructed based on graph-structured data derived from resting-state fMRI data. Formally, we derive a set of graph-structured samples $\{\mathcal{G}^1, \cdots, \mathcal{G}^N\}$ for $N$ participants and $n$ ROIs, where each sample $\mathcal{G}^i = (A^i, X^i, \mathbf{y}^i)$ denotes the functional architecture of the $i$-th objects. In each functional graph of $\mathcal{G}^i$, $A^i \in \{0,1\}^{n \times n}$ is the graph adjacency matrix that represents the functional connection relationship between brain ROIs. $X^i \in \mathbb{R}^{n \times n}$ denotes the node feature matrix that captures functional activity intensity of the brain ROIs, with the corresponding label of $\mathbf{y}^i$ that distinguishes brain disorders or healthy controls (HCs). Our following model aims to learn a map $f(A^i, X^i) \to \mathbf{y}^i$ and discover the biomarkers in the process.

## 3.2 Local-Global Coupling Spiking Graph Transformer

As shown in Fig. 1, our LGC-SGT framework contains the local-global coupling module, the spiking transformer feature extraction block, and the shortcut-enhanced output module with a hybrid loss

function. The model begins by constructing a graph embedding that transforms graph-structured brain ROIs feature matrices and functional connectivity matrices into feature embeddings. Establish a mapping of elementary connection patterns to higher-order pathological features. Specifically, we propose a local-global coupling block that synergistically integrates locally constrained biologically inspired features with globally optimized graph convolutional representations. This integration facilitates precise localization of abnormal brain regions and identification of pathology-relevant biomarkers. Next, we construct the spiking transformer block to effectively capture dependencies among features while strengthening their interactive relationships through dynamic attention weighting. The features are processed through a spike layer, after which the model combines the features before the spike layer with the binarized features from the spike layer to enhance the output used for binary classification between HCs and patients with brain disorders.

**Local-Global Coupling Module.** Building upon the multiple level learning plasticity in the human brain, we propose a local-global coupling (LGC) module designed to simultaneously capture local and global features of brain networks. Specifically, the LGC module integrates two parallel pathways: (1) local pathway learned by unsupervised learning via STDP, and (2) global pathway via supervised learning by backpropagation. This dual-path architecture enables synergistic fusion of neuronal dynamics and network organization.

*Local Pathway.* This pathway mainly consists of three components: an input layer, a reservoir network, and a readout layer. The input layer encodes external signals into spike trains for the reservoir network. The liquid state machine layer comprises spiking neurons with sparse random connectivity, maintaining a biologically plausible excitatory-to-inhibitory ratio of 4:1, which mimics the cortical microcircuit organization [26]. The readout layer, typically implemented as a linear layer, decodes the states of neurons within the reservoir network. The calculation process of LSM can be described as follows:

$$y(t) = f^M((L^M x)(t)), \tag{1}$$

where $x$ is input of LSM, $L^M$ is the nonlinear dynamic system (reservoir network), $f^M$ is the readout function (readout layer), and $y(t)$ is the output of the readout layer at time $t$. The nonlinear dynamic system $L^M$ here is realized with LIF neurons (all spiking neurons in this paper are LIF neurons), given by

$$u^{(t),\,\mathrm{pre}} = \tau u^{(t)} + W x^t, u^{(t)} = u^{(t),\,\mathrm{pre}} \cdot \left(1 - o^{(t)}\right). \tag{2}$$

$$o^{(t)} = \begin{cases} 1 & \text{if } u^{(t),\,\mathrm{pre}} > V_{\mathrm{th}} \\ 0 & \text{otherwise.} \end{cases}, \tag{3}$$

where $u^{(t),\,\mathrm{pre}}$ is the pre-synaptic input at time step $t$, $\tau$ is the leaky factor, $W$ is the weight parameter, $V_{\mathrm{th}}$ is the firing threshold, and $o^{(t)}$ is the spike output. The STDP rule is only used in the local pathway forward propagation, which can be mathematically represented as follows:

$$\Delta W = \begin{cases} A_{\mathrm{LTP}} \cdot e^{-\frac{\Delta t}{\tau_{\mathrm{LTP}}}}, & \text{if } \Delta t > 0 \\[2mm] -A_{\mathrm{LTD}} \cdot e^{\frac{\Delta t}{\tau_{\mathrm{LTD}}}}, & \text{if } \Delta t < 0 \end{cases} \tag{4}$$

where $\Delta W$ represents the change in synaptic weight, $\Delta t$ is the time interval between the pre-synaptic and post-synaptic spikes, $A_{\mathrm{LTP}}$ and $A_{\mathrm{LTD}}$ are control constants of long-term potentiation and long-term depression, respectively, and $\tau_{\mathrm{LTP}}$ and $\tau_{\mathrm{LTD}}$ are time constants.

*Global Pathway.* The local pathways may lead to the loss of original graph structural details. To mitigate this information loss, we introduce a global pathway designed to preserve holistic network attributes. This global pathway employs a spiking graph convolutional network with backpropagation to capture whole-brain dynamics, formally defined by the following equation:

$$H = \tilde{D}^{-\frac{1}{2}} \tilde{A} \tilde{D}^{-\frac{1}{2}} X. \tag{5}$$

$$O_{Global} = SN(BN(Linear(H))) + H, \tag{6}$$

where $\tilde{A} = A + I$ is the adjacent matrix with self-connection, $\tilde{D}$ is the degree matrix of $\tilde{A}$, $H$ is the new node representation with connected aggregation, and $O_{Global}$ is the output of the global pathway. $SN$ represents the spiking layer and $BN$ represents the Batch Norm layer.

**Spiking Transformer Blocks.** We employ a spiking transformer [27] as the backbone network for learning features. The core of the spiking transformer is spiking self-attention (SSA). Specifically, given the feature, it first computes the SSA, given by

$$Q(X) = SN(Linear_q(X)), K(X) = SN(Linear_k(X)), V(X) = SN(Linear_v(X)). \quad (7)$$

$$SSA(X) = MLP(Q(X) \cdot SN(K^T(X) \odot V(X))). \quad (8)$$

After determining the attention, a shortcut message aggregation scheme is applied from the features to the attention, followed by another SNN-enhanced MLP block with the shortcut. The following equations can describe the overall calculation process:

$$Attn^l = SSA(X^l) + X^l, l = 1 \cdots L. \quad (9)$$

$$O^l = MLP(SN(Attn^l)) + Attn^l, l = 1 \cdots L. \quad (10)$$

**Shortcut-enhanced Output Strategy.** Due to the discrete spike firing mechanism employed by SNNs, the step function is non-differentiable. Currently, surrogate gradient methods are widely adopted to address this challenge. However, as the network depth increases, employing surrogate gradients inevitably leads to approximation errors and gradient vanishing. Typically, shortcuts can be introduced to alleviate gradient vanishing and reduce the information loss [28], but this causes a conflict between the real-valued features in intermediate layers and the binarization of SNNs. To address this, researchers have proposed placing spiking activation functions at the beginning of each block to maintain local binarization [29], while enforcing a final spiking layer at the end of the network to achieve global binarization, thus ensuring the network remains purely event-driven. However, binary spike outputs result in information loss. To mitigate the information loss caused by binarization, particularly its impact on classification performance, we propose a shortcut-enhanced output strategy by mixing the features before the spiking layer with their subsequent binarized features for the final output:

$$O_{\text{final}} = O^l \cdot SN_{\text{final}}(O^l), \quad (11)$$

where $O^l$ represents the continuous features before the spiking layer, and $SN_{\text{final}}(O^l)$ converts the continuous features into spiking features. The final output is obtained by multiplying the two, where the spiking signals represent important features, and the real-valued coefficients determine the intensity of the features.

### 3.3 Minimum Error Entropy Criterion

Significant individual differences exist between patients and HCs, and inter-site variability caused by differences in equipment and operational procedures further complicates the data. Traditional loss functions based on cross-entropy (CE) may lack robustness when dealing with such complex data distributions. To address this limitation, we introduce minimum error entropy (MEE) [30] as the extra optimization objective of the model. By minimizing the information entropy of the classification errors, MEE effectively suppresses the interference caused by outliers arising from individual and site differences while reducing reliance on assumptions about specific data distributions.

In cross-entropy loss, the optimization objective essentially measures the difference between two probability distributions, with the optimization performed independently on a point-by-point basis. In contrast, MEE takes into account the statistical dependencies among the error samples. The error information $e$ can be measured by Rényi's entropy [31]

$$H_\alpha(e) = \frac{1}{1 - \alpha} \log V_\alpha(e), \quad (12)$$

where error $e = \hat{y} - y$ between target $\hat{y}$ and model output $y$, $\alpha$ is the order of entropy, usually $\alpha = 2$ for the convenience of calculation, and $V_\alpha(e)$ is the information potential, given by

$$\hat{V}_2(e) = \int \hat{p}^2(e) = \frac{1}{N^2} \sum_{i=1}^N \sum_{j=1}^N G_\sigma(e_i - e_j). \quad (13)$$

$$\hat{p}(x) = \frac{1}{N} \sum_{i=1}^N G_\sigma(x - e_i), \ G_\sigma(e) = \exp\left(-\frac{e^2}{2\sigma^2}\right), \quad (14)$$

where $G_\sigma$ is Gaussian kernel with bandwidth $\sigma$, and $p(:)$ is the probability density function of error $e$. Obviously, minimizing the error entropy criterion $H_2(e)$ is the same as maximum information potential $\hat{V}_2(e)$. Hereby, we define the MEE-Loss as:

$$\mathcal{L}_{MEE} = \hat{H}_2(e) = -\log(\frac{1}{N^2} \sum_{i=1}^{N} \sum_{j=1}^{N} G_\sigma(e_i - e_j)). \tag{15}$$

Then the overall loss can be calculated as follows:

$$\mathcal{L}_{Total} = \mathcal{L}_{CE} + \lambda \mathcal{L}_{MEE}. \tag{16}$$

**Theoretical Analysis**    To validate the robustness of MEE-loss, we conducted a theoretical analysis focusing on gradient perturbations caused by outliers. The CE-loss is defined as follows:

$$\mathcal{L}_{CE} = -\sum_{i=1}^{C} p_i \log q_i, \ \ q_i = \frac{e^{z_i}}{\sum_{j=1}^{C} e^{z_j}}, \ z_i = f(x; \theta). \tag{17}$$

where $p_i$ is the target distribution, $q_i$ is the predict distribution, $f$ is the network mapping, $z_i$ is the network output for category $i$, $x$ is input data, $\theta$ is the learnable network parameter. Next, we compute the gradient:

$$\nabla_\theta \mathcal{L}_{CE} = -\sum_{i=1}^{C} \frac{\partial \mathcal{L}_{CE}}{\partial z_i} \cdot \nabla_\theta z_i = -\sum_{i=1}^{C} (p_i - q_i) \cdot \nabla_\theta z_i. \tag{18}$$

Similarly, the gradient of MEE-loss is given by:

$$\nabla_\theta J_{MEE} = -\log(\frac{1}{N^2} \sum_{i=1}^{N} \sum_{j=1}^{N} G'_\sigma(e_i - e_j) \cdot (\nabla_\theta e_i - \nabla_\theta e_j)). \tag{19}$$

If there are outliers in the input, the corresponding network will produce abnormal outputs. The gradient of CE is linearly related to $z_i$, such that $\nabla_\theta z_i \uparrow$ as $|z_i| \uparrow$. As outliers increase, their influence on parameter updates grows linearly, directly affecting the gradient update. In contrast, for MEE, the Gaussian kernel mapping results in a decrease in the contribution of the kernel function as outliers increase: $G'_\sigma(e_i - e_j) \downarrow$ as $|e_i - e_j| \uparrow$. This leads to the contribution of outliers to the gradient approaching zero.

## 4    Experiments

In this section, we analyze the following aspects to demonstrate the effectiveness of our proposed LGC-SGT model and its dual-perspective biomarker discovery capability.
**Q1.** How does the performance of the model compare to other state-of-the-art baseline models?
**Q2.** Does the model detect abnormal firing rate responses of specific neuronal populations?
**Q3.** Does the model identify the functional connectivity changes in disorder-sensitive regions?

### 4.1    Experimental Settings

**Datasets, Data Preprocessing, and Implementation Details.**    We evaluate the proposed LGC-SGT model using three brain network analysis-related fMRI datasets. (1) The ABIDE dataset [1], which contains 528 patients with autism spectrum disorder (ASD) and 571 HCs. (2) The REST-meta-MDD dataset[2], which contains 848 patients with MDD and 794 HCs. (3) The SRPBS dataset[3], which contains 92 patients with schizophrenia and 92 HCs. The details of datasets, data preprocessing, and experimental implementation can be found in Appendix A.

---

[1]`http://preprocessed-connectomes-project.org/abide/`
[2]`https://rfmri.org/maps`
[3]`https://bicr-resource.atr.jp/srpbsfc/`

**Baseline Models.** The selected baselines correspond to two categories. The first category is ANNs, including generalized graph networks and specialized brain networks. Specifically, these include GCN [32], GAT [33], GIN [34], SIB [35], DIR-GNN [36], ProtGNN [37], BrainGNN [13], IBGNN [38], CI-GNN [39], BrainIB [15], ContrastPool [14], and ALTER [16]. The second category is the SNNs, including SpikingGCN [40], MSG [41], SiGNN [42], and SpikingGT [27].

## 4.2 Performance Comparison (Q1)

**Tenfold Cross-Validation.** Table 1 presents a comparison between the proposed LGC-SGT model and the baseline models. The LGC-SGT significantly outperforms both categories of baseline methods across all three datasets. *(a) Compared to ANNs.* For generalized graph networks, our model demonstrates notable improvements in the accuracy metric, achieving a 6.2% increase on the REST-meta-MDD dataset, a 5.1% increase on the ABIDE dataset, and a 9.2% increase on the SRPBS dataset. For specialized brain networks, we also observe superior performance, with ACC improvements of 0.9%, 2.5%, and 3.2% on REST-meta-MDD, ABIDE, and SRPBS, respectively. *(b) Compared to SNNs.* Our model achieves improvements of 4.9%, 4.3%, and 4.8% on REST-meta-MDD, ABIDE, and SRPBS datasets, respectively. These experimental results indicate that LGC-SGT consistently outperforms the existing baseline methods across all evaluated datasets.

Table 1: Tenfold cross-validation performance with ANNs and SNNs on three datasets (REST-meta-MDD, ABIDE, and SRPBS) (%). The best results are marked in **bold**.

| Category | | Methods | REST-meta-MDD | ABIDE | SRPBS |
|---|---|---|---|---|---|
| ANN | Generalized | GCN [32] | 60.8 ± 1.3 | 64.4 ± 3.8 | 82.7 ± 8.6 |
| | | GAT [33] | 64.7 ± 1.7 | 67.9 ± 3.9 | 84.2 ± 8.1 |
| | | GIN [34] | 65.4 ± 3.2 | 67.9 ± 3.5 | 84.3 ± 8.2 |
| | | SIB [35] | 57.7 ± 3.2 | 62.7 ± 4.4 | 81.1 ± 9.0 |
| | | DIR-GNN [36] | 64.4 ± 1.8 | 67.0 ± 2.7 | 83.9 ± 6.5 |
| | | ProtGNN [37] | 61.0 ± 2.1 | 65.3 ± 3.1 | 84.3 ± 5.0 |
| | Specialized | BrainGNN [13] | 60.2 ± 3.0 | 62.7 ± 2.1 | 82.2 ± 8.5 |
| | | IBGNN [38] | 63.0 ± 2.7 | 64.0 ± 3.1 | 84.8 ± 8.6 |
| | | CI-GNN [39] | 66.8 ± 4.0 | 67.5 ± 3.3 | 87.5 ± 6.8 |
| | | BrainIB [15] | 70.0 ± 2.2 | 70.2 ± 2.0 | 90.3 ± 4.6 |
| | | ContrastPooL [14] | 65.1 ± 1.9 | 68.6 ± 2.7 | 89.2 ± 3.4 |
| | | ALTER [16] | 65.8 ± 3.3 | 70.5 ± 1.4 | 89.5 ± 5.9 |
| SNN | Generalized | SpikeGCN [40] | 66.0 ± 2.1 | 68.7 ± 1.6 | 86.5 ± 2.5 |
| | | MSG [41] | 60.4 ± 2.4 | 62.9 ± 3.7 | 81.9 ± 2.7 |
| | | SiGNN [42] | 66.2 ± 2.5 | 69.1 ± 1.8 | 87.6 ± 7.2 |
| | | SpikeGT [27] | 61.8 ± 3.5 | 65.8 ± 1.4 | 88.7 ± 3.6 |
| | Specialized | LGC-SGT (Ours) | **70.9 ± 1.7** | **73.0 ± 2.6** | **93.5 ± 3.0** |

Table 2: Leave-one-site-out cross-validation performance on REST-meta-MDD and ABIDE dataset (%).

| Dataset | Methods | S1 | S2 | S3 | S4 | S5 | S6 | S7 | S8 | S9 | S10 | S11 | S12 | S13 | S14 | S15 | S16 | S17 | Mean |
|---|---|---|---|---|---|---|---|---|---|---|---|---|---|---|---|---|---|---|---|
| REST-meta-MDD | CI-GNN | 63.3 | **83.0** | 85.2 | 76.3 | 70.0 | **68.8** | 75.4 | 73.2 | 72.2 | 68.3 | 81.1 | 62.3 | 67.3 | **63.2** | 68.3 | **75.2** | 64.4 | 71.6 |
| | BrainIB | 63.3 | 73.0 | **85.4** | **77.8** | 71.3 | **68.8** | 73.2 | 75.7 | **80.6** | 72.0 | **82.1** | **67.1** | **69.4** | **63.2** | 70.1 | 71.1 | 68.9 | 72.5 |
| | SpikeGT | 55.5 | 56.7 | 75.6 | 62.5 | 70.1 | 62.5 | 70.4 | 62.2 | 61.1 | 65.6 | 73.1 | 57.3 | 63.3 | 58.1 | 63.9 | 68.4 | 64.4 | 64.2 |
| | LGC-SGT | **63.7** | 70.0 | **85.4** | **77.8** | **74.7** | 68.4 | **76.1** | **78.4** | **80.6** | **75.3** | **82.1** | 62.2 | **69.4** | 60.4 | **70.8** | 68.9 | **78.9** | **73.1** |
| ABIDE | CI-GNN | 83.3 | 71.1 | 67.4 | 67.8 | 71.7 | 64.3 | **67.9** | 77.8 | **78.1** | 69.2 | **83.3** | **75.0** | 64.2 | 63.4 | 61.2 | 66.3 | 75.1 | 71.0 |
| | BrainIB | 83.3 | 71.1 | **72.7** | **73.4** | 66.7 | **70.1** | **67.9** | 77.8 | 66.7 | **83.3** | 75.0 | **75.0** | 65.3 | 74.7 | 64.8 | 73.3 | 82.1 | 73.1 |
| | SpikeGT | 79.2 | 57.9 | **72.7** | 67.2 | 61.4 | 69.0 | 53.6 | 72.2 | 56.1 | 70.0 | 72.2 | 70.0 | 69.4 | 68.7 | 63.5 | 74.3 | 78.6 | 68.0 |
| | LGC-SGT | **87.5** | **76.3** | 70.9 | **73.4** | **73.7** | **70.1** | 64.3 | **80.6** | 70.2 | 73.3 | 77.8 | 72.5 | **73.5** | **74.8** | **67.6** | **75.3** | **85.7** | **74.6** |

**Leave-One-Site-Out Cross-Validation.** To further assess the performance of our model, we conduct leave-one-site-out cross-validation. ABIDE and REST-meta-MDD datasets contain 17

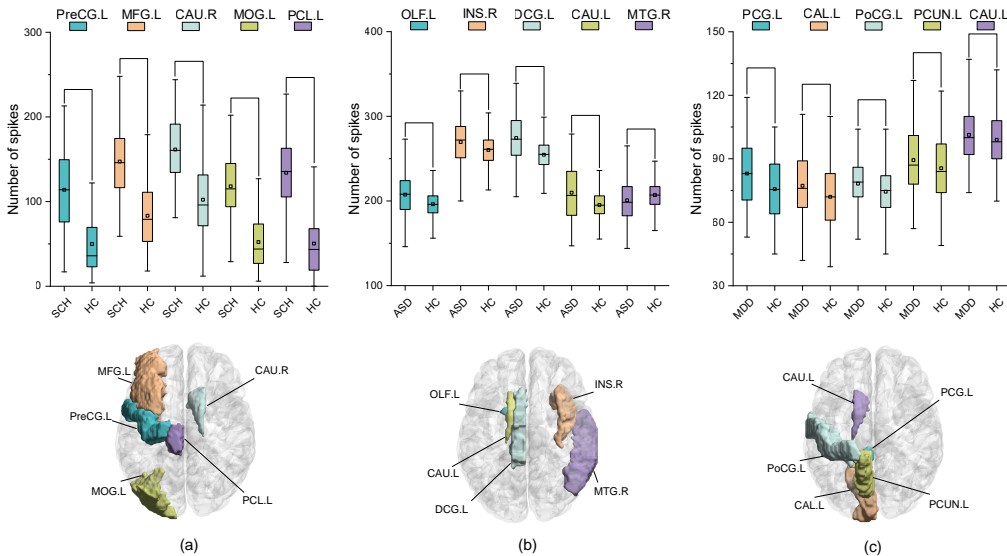

Figure 2: Comparison of brain region activity of HCs and patients on schizophrenia, ASD, and MDD. The top 5 different regions are shown. The two-sample *t*-test with Welch's correction was employed to evaluate the significance of intergroup differences. Statistical significance was defined as $p < 0.05$. (a) Schizophrenia. (b) ASD. (c) MDD.

independent sites. In detail, we divide each dataset into the training set (16 sites out of 17 sites) to train the model and the testing set (remaining site out of 17 sites) for testing the model. We compared our model with CI-GNN, BrainIB, and SpikeGT. CI-GNN and BrainIB were chosen due to their performance better in tenfold cross-validation compared to other models, while SpikeGT represents a typical transformer-based SNN model. The experimental results are summarized in Table 2. Compared to the baselines, our model achieved the highest site-average accuracy, reaching 73.1% and 74.6% on REST-meta-MDD and ABIDE datasets, respectively. Moreover, it demonstrates significant improvements across most sites, indicating that the strength of our model lies in its ability to generalize well across completely different multi-site datasets despite site-specific differences.

**Ablation Study.** To rigorously evaluate the contribution of each proposed component to our model's performance, we conduct comprehensive ablation studies across three benchmark datasets. The experimental design systematically isolates three critical elements: (i) the MEE-loss function, (ii) the LGC module, and (iii) the shortcut-enhanced output strategy (SEO). As shown in Table 3, the LGC-SGT achieves the highest classification accuracy on three brain disorders classification tasks, demonstrating the effectiveness of the proposed modules.

Table 3: Ablation study for the modules we design.

| Dataset | Method | Accuracy |
|---|---|---|
| REST-meta-MDD | Baseline | **70.94%** |
| | w/o MEE | 70.77% |
| | w/o LGC | 69.99% |
| | w/o SEO | 69.83% |
| ABIDE | Baseline | **73.03%** |
| | w/o MEE | 72.91% |
| | w/o LGC | 72.55% |
| | w/o SEO | 71.73% |
| SRPBS | Baseline | **93.51%** |
| | w/o MEE | 90.50% |
| | w/o LGC | 89.77% |
| | w/o SEO | 89.74% |

## 5 In-depth Analysis of brain neuronal population dynamics (Q2)

To analyze the neuronal population dynamics of our model, we conducted rigorous statistical comparisons. The results show that the differences in spiking activity between patients with brain disorders and HCs are statistically significant. Fig. 2 illustrates the five ROIs exhibiting maximal inter-group differences for each disorder, accompanied by their neuroanatomical schematics. *Schizophrenia*: Patients demonstrated aberrant activity in several brain regions, including the PreCG.L, MFG.L, PCL.L, CAU.R, and MOG.L. These findings align with prior studies [43–45] that abnormal activity in

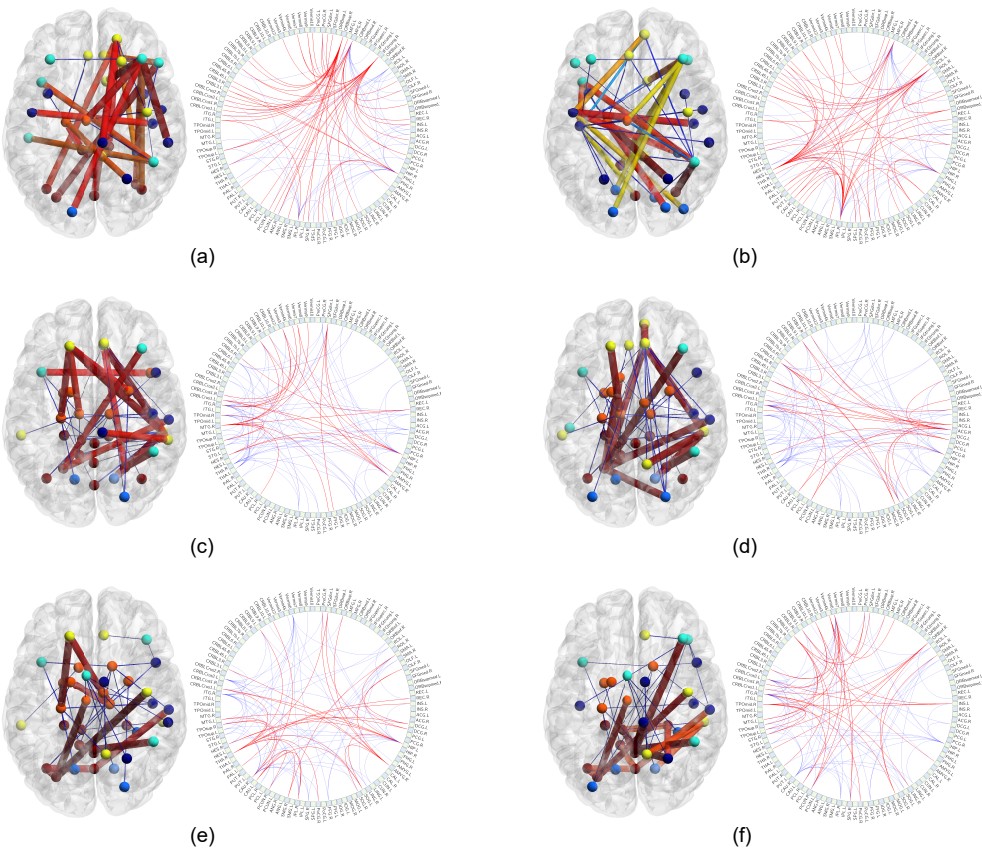

Figure 3: Comparison of brain functional connections of HCs and patients with schizophrenia. In the anatomical brain map, the thickness of the connections represents the strength of functional connectivity; in the circular layout chord diagram, blue lines indicate shared functional connections between the patient and HCs, while red lines represent connections unique to either the patient or HCs. (a) Patients with schizophrenia. (b) HCs of schizophrenia. (c) Patients with ASD. (d) HCs of ASD. (e) Patients with MDD. (f) HCs of MDD.

these regions serves as a potential cause of schizophrenia. These regions are primarily involved in the maintenance of working memory, conflict monitoring, and the regulation of goal-directed behavior. *ASD*: significant activity differences emerge in the OLF.L, INS.R, DCG.L, CAU.L, and MTG.R. Supporting literature [46–48] further suggests that abnormalities in these regions are closely related to the core symptoms characteristic of ASD. These regions are responsible for detecting internal sensory signals and coordinating the allocation of attentional resources. *MDD*: patients showed significant activity alterations in the PCG.L, CAL.L, PoCG.L, PCUN.L, and CAU.L. Consistent with these observations, previous studies [49, 50] have confirmed that abnormal activity in these brain regions may underlie the development of major depressive disorder. Abnormalities in these brain regions may lead to a bias in the processing of negative information.

Comparative analysis across three brain disorders revealed distinct neural activity patterns of aberrant brain regions when contrasted with HCs. These results indicate that our model effectively simulates brain activity and identifies regions exhibiting abnormal neural activity in patients compared to HCs. Clinical and neuroimaging results also validate the rationale of the biomarkers we discovered. The detailed experimental setting can be found in Appendix B.1.

## 6 In-depth Analysis of brain functional connectivity (Q3)

To analyze the functional pattern under different brain disorders, we visualize functional connectivity. The results show that the functional connectivity between patients with brain disorders and HCs

is significantly different. As shown in Fig. 3, our model learned distinct connectivity patterns for schizophrenia patients and HCs, and obtained some functional connections that could distinguish patients from HCs. ***Schizophrenia***: functional connectivity was significantly increased between the MFG.R and IFGtriang.R (frontoparietal network). This finding is consistent with previous studies [51] reporting that the connectivity between the sensorimotor network and the frontoparietal network is enhanced in schizophrenia. ***ASD***: the increased connectivity pointing to the HIP.L region, which aligns with prior research indicating functional abnormalities of the hippocampus in ASD [52]. ***MDD***: connectivity between the THA.R and HIP.L has been increased, consistent with findings from [53] that connectivity change in these regions is part of the pathophysiology of MDD.

The experimental results demonstrate that our model accurately reproduces the abnormal functional connectivity patterns observed clinically, effectively simulates the physiological mechanisms underlying brain functional connectivity, and reveals distinct connectivity patterns characteristic of brain disorders. Clinical and neuroimaging results also validate the rationale of the biomarkers we discovered. The detail of the experiment is provided in the Appendix B.2.

## 7    Conclusion and Discussions

**Conclusion.**    In summary, we propose the LGC-SGT model that jointly models inter-regional connectivity discrepancies and neuronal population firing rate deviations, enabling a dual-perspective analysis of neuropathology. By integrating a shortcut-enhanced output strategy with a hybrid loss function, our framework achieves robust decision boundaries for brain disorder diagnosis. Extensive experiments on three brain disorder datasets demonstrate that the LGC-SGT model consistently outperforms existing graph learning methods, successfully identifying dual-perspective biomarkers from both inter-regional connectivity and neuronal firing rate abnormalities. This approach offers novel insights for clinical research and therapeutic strategies.

**Limitations.**    Although our current work effectively simulates brain functional states using resting-state fMRI data, incorporating multimodal data, such as diffusion tensor imaging (DTI) for structural connectivity and electroencephalogram (EEG) for neural oscillations, would provide additional anatomical constraints and temporal resolution, enabling more accurate brain modeling grounded in biological and clinical relevance.

## Acknowledgments

This work is supported by the National Natural Science Foundation of China (Grant No. 62436005, U21A20485, 62306274), China Postdoctoral Science Foundation under Grant Number (No. GZB20250394), Young Talent Fund of Xi'an Association for Science and Technology (0959202513037).

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

# A  Datasets and Implementation Details

## A.1  Datasets and Data Preprocessing

In the construction of brain connectomes, we first preprocessed the raw resting-state fMRI data using Statistical Parametric Mapping (SPM) software, including slice timing correction and head motion correction. Subsequently, the resting-state fMRI images were normalized to a standard space using deformation parameters that map the fMRI images to the Montreal Neurological Institute (MNI) template. In addition, a Gaussian filter with a half maximum width of 6 mm is used to smooth the functional images. The resulting fMRI images are filtered with a bandpass filter (0.01–0.08 Hz). After preprocessing, the average time series for each participant is extracted from each ROIs using the AAL atlas, which comprises 90 cerebral regions and 26 cerebellar regions. Then, functional connectivity was computed between all ROIs pairs using Pearson's correlation coefficient, followed by Fisher's r-to-z transformation, resulting in a $116 \times 116$ symmetric matrix representing the brain network. Their demographic information is summarized in Table 4.

Table 4: Description of datasets. Demographic and clinical characteristics.

| Characteristic | ABIDE | | Rest-meta-MDD | | SRPBS | |
|---|---|---|---|---|---|---|
| | ASD | HC | MDD | HC | Schizophrenia | HC |
| Sample Size | 528 | 571 | 828 | 776 | 92 | 92 |
| Age | $17.0 \pm 8.4$ | $17.1 \pm 7.7$ | $34.3 \pm 11.5$ | $34.4 \pm 13.0$ | $39.6 \pm 10.4$ | $38.0 \pm 12.4$ |
| Gender(M/F) | 464/64 | 471/100 | 301/527 | 318/458 | 47/45 | 60/32 |

## A.2  Implementation Details

In the proposed LGC-SGT model, we had the following hyperparameters and the range of them in Table 5. And all experiments were performed on a single NVIDIA RTX 3090 GPU.

Table 5: Ranges of different hyperparameters.

| Parameter | Rest-meta-MDD | ABIDE | SRPBS |
|---|---|---|---|
| number of spiking transformer blocks | 3 | 2 | 1 |
| timesteps | 8 | 2 | 4 |
| lambda | 0.1 | 0.2 | 0.2 |
| MEE kernel | 0.5 | 0.1 | 0.2 |
| batch size | 32 | | |
| learning rate | $1e-4$ | | |
| weight decay | $1e-4$ | | |
| attention heads | 2 | | |
| embedding dimension | 116 | | |

# B  Two Perspective Biomarker Discovery

The differences in the association mechanisms between abnormal activity in brain regions and abnormal functional connectivity in pathological contexts primarily reflect their impact on different levels of neural networks: the former focuses more on the activity levels of local brain regions, while the latter emphasizes the functional connectivity relationships between brain regions.

## B.1  Brain Neuronal Population Dynamics

The analysis employs the Automated Anatomical Labeling (AAL) atlas as the criteria for brain region division, establishing a precise one-to-one mapping between 116 spiking neurons and neuroanatomical regions (90 cerebral and 26 cerebellar).

## B.2 Brain Functional Connectivity

To analyze brain functional connectivity under different brain disorders, we performed separate STDP training of the model's local pathways using samples from each disorder group and the HCs. To further investigate disorder-specific differences in functional connectivity patterns, we conducted the following analyses on the trained networks for each group: first, we established a mapping between the model's reservoir layer neurons and specific brain regions; second, we constructed functional connectivity maps based on the synaptic weight matrices obtained after training, quantifying the interaction strength between different brain regions; finally, by comparing network characteristics between patient and HCs, we identified abnormal connectivity patterns specific to each disorder, which may help elucidate the underlying neural circuit mechanisms of brain disorders. For visualization, we selected the top 100 edges with the highest weights along with their connected nodes.

