# OpenReview forum: "Local-Global Coupling Spiking Graph Transformer  for Brain Disorders Diagnosis from Two Perspectives"
_NeurIPS.cc/2025/Conference — NeurIPS 2025 poster_

### Official Review · Reviewer_us7b · 2025-06-14

**Clarity:** 3
**Significance:** 3
**Originality:** 2
**Rating:** 4
**Confidence:** 3

**Summary:**

The paper proposed a model called Local-Global Coupling Spiking Graph Transformer (LGC-SGT) to predict brain disorders or healthy controls from functional connections and neuron activity data. The proposed model consists of a biological plausible part (learned by reservoir computing) for capturing the local pathway, and a graph neural network part for capturing the global pathway. The experiments show that the LGC-SGT achieves superior results compared to its competitors.

**Questions:**

1. What is the neural population size of each regions in the LSM? Is the size of the neural population has great impact on the results?
2. How to set the input to the LSM (P4 L142, the vector x)? The model takes fMRI data as its inputs. Does the LSM transform the input fMRI data into spikes?
3.What is the benefits of using spiking-based network (LSM and spiking transformer)? Why not using continuous-value reservoir computing to handle the input fMRI data and using the continuous transformer. The utilization of spiking-based computing is for energy saving in general, but the proposed model is not large and seems no necessary for energy saving. Additionally, the fMRI data lack single-neuron resolution, so the spike trains in the model do not directly correspond to biological neurons.
4. What is the implementation details of the STDP used in the paper for LSM?
5. Can the weight connections learned by the model in the LSM be consistent with that in the brain?

**Ethical Concerns:**

["NO or VERY MINOR ethics concerns only"]

**Final Justification:**

The author’s rebuttal addresses most of my concerns. However, I remain unconvinced about the suitability of spiking neural networks (SNNs) for disorder diagnosis. The authors could further elaborate on the advantages of SNNs beyond energy efficiency—such as biological plausibility and explainability to strengthen their argument. Given the current manuscript and rebuttal, I maintain my original score.

**Limitations:**

1. The use of spiking-based computing seems unnecessary.
2. All the components used in the paper are not original, limiting the contribution and originality of the paper.

**Paper Formatting Concerns:**

There is no paper formatting concerns.

**Quality:**

3

**Strengths And Weaknesses:**

### Strength:
1. The paper is clearly written and expressed.
2. The proposed method show good performance on the brain disorder prediction task.
3. The ablation study make it convincing that each component of the proposed method is useful.

### Weakness:
1. Using spiking-based model for biological plausibility seems unnecessary. Why not use continuous reservoir computing and continuous transformer, which may has better performance?
2. The theoretical analysis of the MEE is not convincing. For an outlier input, the gradient if the MEE loss is vanishing due to the Gaussian kernel, however, the gradient of the CE loss is still no vanishing.

---

> ### Author Rebuttal · Authors · 2025-07-30
>
> We are grateful for the reviewers’ insightful comments. Please find our detailed responses below. We trust they resolve the raised concerns and welcome any further questions.
>
> **W1. About why not use continuous reservoir computing and continuous transformer.**
>
> A core aspect of this study is to utilize the STDP learning rule to construct representations of inter-regional brain connections. The essence of the STDP rule is to adjust synaptic weights based on the precise firing times of spiking neurons. Continuous reservoir computing models typically do not produce spiking outputs directly, as their state updates are continuous, which is incompatible with the discrete, temporal spike events relied upon by the STDP rule. To fully leverage the advantages of spiking neural networks in terms of energy efficiency and temporal information processing, and to maintain the unity of the entire model architecture, we employ a spiking transformer that can receive spike inputs and process information in a spiking manner, forming an end-to-end spiking processing process.
>
> ---
>
> **W2. About the theoretical analysis of the MEE.**
>
> We thank the reviewer for this insightful comment. It should be clarified that the MEE loss introduced in our method is an auxiliary regularization term, whose core purpose is to enhance the consistency of the overall model output. Specifically, the CE loss is responsible for optimizing the classification accuracy of individual samples, while the MEE loss responds to the weakening of model output consistency by measuring and minimizing the differences in sample errors within the batch. When the influence of outliers increases, the MEE loss value will also increase accordingly. Through joint optimization with the CE loss, not only does it focus on individual fitting, but also on the consistency of the overall prediction behavior, which helps to smooth the learning process and reduce the impact of outliers.
>
> To ensure that our proposed method can effectively suppress outliers, we conducted experiments on the SRPBS dataset with Gaussian noise and salt-and-pepper noise (intensity is 0.1). The experimental results show that without MEE, the model is more susceptible to the influence of noise. It indicates that MEE can effectively reduce the impact of noise.
>
> | Accuracy (\%)   | w/o  MEE |Ours |
> |:-------|:-------:|:-------:|
> | Gaussian noise         | 85.3(7.8)   |  86.5(7.7)  |
> | Salt-and-pepper noise  | 82.7(9.0)   | 83.8(10.7)  |
>
> ---
>
> **Q1. The neural population size of each regions in the LSM.**
>
> Thanks for your insightful comment. We apply different neural populations settings in the LSM as follows:
> | Accuracy (\%)   | 2 |4 | 8 |16 |
> |:-------|:-------:|:-------:|:-------:|:-------:|
> | SRPBS         |  91.9(4.7) |92.4(4.8) | 93.2(3.8) | 93.1(3.6)  |
>
> The experimental results in the table show that the performance increases first and remains stable as the number of neural population sizes increases.
>
> ---
>
> **Q2. About the input to LSM.**
>
> Thanks for the question. Yes, the LSM takes the input fMRI data as the membrane potential of the spiking neurons, which then generate spikes.
> In detail, we first preprocessed the raw resting-state fMRI data using Statistical Parametric Mapping (SPM) software. Then, we mapped the preprocessed data to brain regions using the AAL atlas and extracted the average features for each region. Finally, we computed the functional connectivity matrix and average brain region activity intensity between all region pairs using Pearson's correlation coefficient. The input to the LSM is each regional average activity intensity.
>
> ---
>
> **Q3. The benefit of using spiking-based model.**
>
> Thanks for the insightful comment. There are three reasons: firstly, one of the advantages of spiking-based models is indeed energy efficiency as what you said. The proposed LGC-SGT decreases about 70\% compared to the ANNs according to the energy estimation (BNT [1]/BIOBGT [2]/BrainOOD [3] 0.20/0.26/0.25 mJ vs. ours 0.035 mJ). Secondly, our model achieves higher performance (BrainOOD/BrainIB/ALTER 90.2/90.3/89.5\% vs. ours 93.5\% accuracy) by exploiting the neuronal dynamics modeling capability of the spiking-based model, combined with the designed LGC-SGT framework. Finally, we dig the advantage of the biological plausibility of the spiking-based model to provide a new perspective on neural dynamics changes for the disorder diagnosis problem. Combined with the functional connectivity, thus could form the two perspectives from both connectivity and neuronal dynamics for the brain disorder diagnosis problem.
>
> ---
>
> **Q4. About the STDP implementation.**
>
> Thanks for pointing that out. STDP adjusts synaptic strength based on the precise temporal order of spikes emitted by pre- and post-synaptic neurons. This usually manifests as long-term potentiation (LTP) and long-term depression (LTD). Specifically, if a pre-synaptic neuron fires shortly before a post-synaptic neuron, the synapse is typically strengthened; conversely, if the post-synaptic neuron fires first, the synapse is weakened.
> This timing-dependent adjustment allows neural networks to learn connection weights in an unsupervised and biologically plausible manner. The STDP rule can be mathematically represented as follows:
> $$
> \Delta W =
> \begin{cases}
> A_{\mathrm{LTP}} \cdot e^{-\frac{\Delta t}{\tau_{\mathrm{LTP}}}}, & \text{if } \Delta t > 0 \\\\
> -A_{\mathrm{LTD}} \cdot e^{\frac{\Delta t}{\tau_{\mathrm{LTD}}}}, & \text{if } \Delta t < 0
> \end{cases}
> $$
> where $\Delta W$ represents the change in synaptic weight, $\Delta t$ is the time interval between the pre-synaptic and post-synaptic spikes, $A_{\mathrm{LTP}} = 0.1$ and $A_{\mathrm{LTD}} = 0.12$ are positive constants that control the magnitude of LTP and LTD, respectively, and $\tau_{\mathrm{LTP}} = 20$ and $\tau_{\mathrm{LTD}} = 20$ are time constants.
> The LSM is built based on spiking neurons, and the local synaptic learning by STDP rules in LSM proceeds weight updating. The update process of the LSM module and the global BP spiking-based transformer architecture based on local STDP learning rules is iteratively learned. We will supplement the detailed explanations in our upcoming revision.
>
> ---
>
> **Q4. About whether the weight connections learned by the model in LSM are consistent with those in the brain.**
>
> The weight connections learned by the model in the LSM are not completely consistent with those in the brain. To quantify this, we computed the cosine similarity between the weights learned by the LSM and the average connectivity weights of the SRPBS dataset, which yielded a result of 0.446. This moderate similarity suggests that while there is some correspondence between the learned weights and the original brain connectivity patterns, they are not identical. Therefore, while the LSM captures certain aspects of brain connectivity, the learned weights represent a simplified approximation that primarily retains the important connections in the brain network while filtering out redundant or secondary connections.
>
> We will also provide detailed explanations in our upcoming revision.
>
> ---
>
> **About the contribution.**
>
> Thank you for your valuable feedback. We would like to emphasize that our local-global framework is novel and represents more than a simple combination of existing components. ur motivation is to effectively model neuronal firing dynamics and functional connectivity patterns to achieve more accurate brain disorder diagnosis.
> On one hand, the local-global framework is specifically designed for brain disorder diagnosis, taking into account the distinct properties of regional neuronal dynamics and functional connectivities. On the other hand, while our local component is inspired by the LSM architecture and the global component is built upon the transformer architecture, these models are not identical to their original counterparts. Furthermore, the shortcut enhancement module is a newly designed component that can enhance the intensity of model information output and facilitate faster training convergence.
>
> ---
>
> **Reference**
>
> [1] Brain Network Transformer. NeurIPS 2022.
>
> [2] Biologically Plausible Brain Graph Transformer. ICLR 2025.
>
> [3] BrainOOD: Out-of-distribution Generalizable Brain Network Analysis. ICLR 2025.

---

> > ### Comment · Reviewer_us7b · 2025-08-04
> > **Thank you for the rebuttal**
> >
> > The author’s rebuttal addresses most of my concerns. However, I remain unconvinced about the suitability of spiking neural networks (SNNs) for disorder diagnosis. The authors could further elaborate on the advantages of SNNs beyond energy efficiency—such as biological plausibility and explainability to strengthen their argument. Given the current manuscript and rebuttal, I maintain my original score.

---

> > > ### Author Response · Authors · 2025-08-04
> > >
> > > Thank you for your feedback. We appreciate your insightful comments and would like to respond to your questions as follows.
> > >
> > > **Biological plausibility and explainability.**
> > >
> > > SNNs offer distinct advantages in biological interpretability. Spiking neurons communicate through discrete spiking events, a firing pattern that closely resembles the electrophysiological behavior of real neurons in the biological brain. This enables a more accurate simulation of regional brain activity compared to traditional artificial neural networks (ANNs). Unlike ANNs, which rely on continuous and abstract activation values, SNNs encode information in spike trains that capture key population-level neural properties such as firing rate, spike timing, neuronal synchrony, and spatiotemporal activation patterns. This provides a more biologically grounded modeling framework for investigating abnormal neural dynamics and pathological discharge patterns in neurological and neuropsychiatric disorders.
> > >
> > > Furthermore, in this study, we employ the LSM combined with STDP, a biologically plausible synaptic plasticity mechanism, to model functional brain connectivity. STDP adjusts synaptic weights based on the temporal order and interval between presynaptic and postsynaptic spikes, effectively implementing a form of Hebbian learning. This mechanism allows the network to simulate neurobiological processes such as synaptic strengthening or weakening during learning and development. Importantly, it can also capture pathological changes in connectivity observed in brain disorders, such as impaired synaptic pruning or the emergence of hyperexcitable circuits.
> > >
> > > By jointly modeling spiking dynamics and adaptive connectivity, SNNs achieve a higher degree of similarity to real neural circuits in both structural organization and functional dynamics. This integrated approach provides a multi-faceted and interpretable framework for probing the network-level mechanisms underlying brain disorders. The incorporation of biologically realistic principles not only improves the model’s interpretability at the neural circuit level but also enhances the reliability of brain disorder diagnosis.
> > >
> > > We sincerely thank you for your efforts in reviewing our paper. We hope we have resolved all the concerns, and we will deeply appreciate it if you could reconsider the score accordingly. We are always willing to address any of your further concerns.

---

### Official Review · Reviewer_5jm3 · 2025-06-22

**Clarity:** 2
**Significance:** 3
**Originality:** 3
**Rating:** 4
**Confidence:** 2

**Summary:**

The paper proposes a novel Local-Global Coupling Spiking Graph Transformer (LGC-SGT) model for diagnosing brain disorders from two perspectives: inter-regional connectivity and neuronal population firing rates within brain regions. The model integrates a spike-based transformer with a local-global coupling module and a shortcut-enhanced output strategy to capture both global topology shifts and local firing rate dynamics.

**Questions:**

1.	The author should provide more detailed explanations about STDP implementation (Lines 132-147) to clarify the unsupervised learning process in the Local-Global Coupling Module.
2.	How were the key hyperparameters (number of spiking transformer blocks, time steps, lambda, and MEE kernel) determined? Was any ablation study conducted to verify their optimality across different datasets?
3.	It is suggested to clarify the definition of $SN_{final}$ in Eq. (10) and discuss the advantages/disadvantages of mixing features before the spiking layer with binarized features.
4.	Why weren't more recent SNN-based graph learning methods included in the comparisons, given that most baselines are ANN-based baselines?

**Ethical Concerns:**

["NO or VERY MINOR ethics concerns only"]

**Final Justification:**

I believe most of my concerns have been well-addressed. However, upon a comprehensive analysis and careful consideration of other reviewers' comments, it has become apparent during our discussion that the current paper still has significant room for improvement. The numerous supplementary experiments also need to be further detailed and refined within the paper to enhance its overall quality. Despite this, I have decided to increase my score, though with a low confidence level.

**Limitations:**

Yes

**Paper Formatting Concerns:**

The paper generally follows the NeurIPS 2025 Paper Formatting Instructions.

**Quality:**

2

**Strengths And Weaknesses:**

Strengths:

1.The idea of combining a spike-based transformer with local-global coupling to jointly model inter-regional connectivity and neuronal population firing rates is interesting.

2.The ability to extract interpretable neurobiological biomarkers from both inter-regional connectivity and neuronal firing rate abnormalities could offer valuable insights for clinical research and therapeutic strategies.

Weaknesses:

1.Some technical details in the manuscript are not clearly explained. For instance, in Line 132 of the Local-Global Coupling Module, the authors mention that the local pathway is learned by unsupervised learning via STDP, but they fail to provide a detailed explanation of STDP and its specific implementation (including the descriptions in Lines 135-147), which may cause confusion for readers.

2.The choice of hyperparameters, such as the number of spiking transformer blocks, time steps, lambda, and MEE kernel, is not justified. It is unclear how these hyperparameters were determined and whether they are optimal for different datasets.

3.In Eq. (10), the exact definition of $SN_{final}$ remains unclear, as the authors do not provide sufficient explanation. Additionally, the benefits and potential drawbacks of mixing the features before the spiking layer with the binarized features for the final output are not thoroughly discussed. A more detailed analysis on this aspect should be clearly discussed.

4.The proposed method lacks comprehensive comparisons with existing approaches. Since this work is an SNN-based method, it is insufficient to compare primarily with ANN-based methods (all 12 baselines in the paper). Although two SNN-based baselines are included (e.g., SpikeGCN in 2022), they are relatively outdated. The authors should compare with more recent SNN-based graph learning methods, such as those in [1]–[4].

[1] Dynamic spiking graph neural networks[C]//Proceedings of the AAAI Conference on Artificial Intelligence. 2024.

[2] Continuous spiking graph neural networks[J]. arXiv preprint arXiv:2404.01897, 2024.

[3] SpikeGraphormer: A high-performance graph transformer with spiking graph attention[J]. arXiv preprint arXiv:2403.15480, 2024.

[4] SiGNN: A spike-induced graph neural network for dynamic graph representation learning[J]. Pattern Recognition, 2025.

---

> ### Author Rebuttal · Authors · 2025-07-30
>
> We are grateful for the reviewers’ insightful comments. Please find our detailed responses below. We trust they resolve the raised concerns and welcome any further questions.
>
> **Q1. About the STDP implementation.**
>
> We thank the insightful feedback. The unsupervised Spike-Timing-Dependent Plasticity (STDP) rule adjusts synaptic strength based on the precise temporal order of spikes emitted by pre- and post-synaptic neurons. This usually manifests as long-term potentiation (LTP) and long-term depression (LTD). Specifically, if a pre-synaptic neuron fires shortly before a post-synaptic neuron, the synapse is typically strengthened; conversely, if the post-synaptic neuron fires first, the synapse is weakened. This timing-dependent adjustment allows neural networks to learn connection weights in an unsupervised and biologically plausible manner. The STDP rule can be mathematically  represented as follows:
>
> $$
> \Delta W =
> \begin{cases}
> A_{\mathrm{LTP}} \cdot e^{-\frac{\Delta t}{\tau_{\mathrm{LTP}}}}, & \text{if } \Delta t > 0 \\\\
> -A_{\mathrm{LTD}} \cdot e^{\frac{\Delta t}{\tau_{\mathrm{LTD}}}}, & \text{if } \Delta t < 0
> \end{cases}
> $$
>
> where $\Delta W$ represents the change in synaptic weight, $\Delta t$ is the time interval between the pre-synaptic and post-synaptic spikes, $A_{\mathrm{LTP}} = 0.1$ and $A_{\mathrm{LTD}} = 0.12$ are positive constants that control the magnitude of LTP and LTD, respectively, and $\tau_{\mathrm{LTP}} = 20$ and $\tau_{\mathrm{LTD}} = 20$ are time constants.
> During the training phase (epoch=100), we employed a hybrid training strategy: STDP training was executed during the forward propagation process, optimizing only the internal connection weights of the LSM. This was followed by BP training, which optimized the global parameters of the entire network model, excluding the internal connection weights of the LSM.
>
> ---
>
> **Q2.About the choice of hyperparameters.**
>
> Thanks for your insightful suggestions. All hyperparameters were determined through grid search in our model.  The specific ablation experiments are as follows.
>
> | Spiking transformer blocks  | Rest-meta-MDD | ABIDE | SRPBS |
> |:-------:|:-------:|:-------:|:-------:|
> |1   | 69.2(2.6)    | 72.2(2.7)   |93.5(3.0)   |
> |2   | 69.5(3.1)    |73.0(2.6)     | 92.9(4.4)  |
> |3   | 70.9(1.7)    |  72.1(1.8)   | 92.5(3.7)  |
> |4   | 70.5(4.3)    |  72.6(1.8)   | 92.5(3.7) |
>
> | Timesteps  | Rest-meta-MDD | ABIDE | SRPBS |
> |:-------:|:-------:|:-------:|:-------:|
> |2   | 68.6(2.0)   | 73.0(2.6)  |91.9(3.2) |
> |4   | 70.1(2.3)   |72.8(2.1)   | 93.5(3.0) |
> |8  | 70.9(1.7)   |  72.3(3.5)   | 93.1(4.4) |
>
> | Lambda  | Rest-meta-MDD | ABIDE | SRPBS |
> |:-------:|:-------:|:-------:|:-------:|
> |0.1   | 70.9(1.7)   | 72.5(1.7)  |91.9(3.8)  |
> |0.2   | 69.4(2.5)   |73.0(2.6)   | 93.5(3.0) |
> |0.3   | 69.6(3.3)   | 71.0(3.9)  | 92.4(3.7) |
>
> | MEE kernel  | Rest-meta-MDD | ABIDE | SRPBS |
> |:-------:|:-------:|:-------:|:-------:|
> |0.1    | 68.9(3.5) | 73.0(2.6) | 91.9(3.8) |
> |0.2    | 68.8(2.9) | 72.7(2.2) | 93.5(3.0) |
> |0.3    | 69.2(3.2) | 71.8(1.7) | 93.0(4.4) |
> |0.4    | 70.3(4.1) | 71.5(1.8) | 91.9(3.8) |
> |0.5    | 70.9(1.7) | 70.5(2.0) | 91.9(3.8) |
>
> The specific parameter values for the selection of each dataset are as follows.
> | Dataset  | Rest-meta-MDD | ABIDE | SRPBS |
> |:-------|:-------:|:-------:|:-------:|
> |  Spiking transformer blocks | 3 | 2 | 1 |
> |  Timesteps                  | 8 | 2 | 4 |
> |  Lambda                     | 0.1 | 0.2 | 0.2 |
> |  MEE kernel                  | 0.5 | 0.1 |0.2 |
>
> ---
>
> **Q3. Clarified definition in Eq.10.**
>
> Eq. (10) defines the shortcut-enhanced output strategy, which addresses the information loss caused by binarization through a hybrid representation of real-valued and spiking features. The specific definition is as follows:  $O^{l}$ represents the continuous features before the spiking layer, and $SN_{\mathrm{final}}(O^{l})$ converts the continuous features into spiking features. The final output is obtained by multiplying the two, where the spiking signals represent important features, and the real-valued coefficients determine the intensity of the features.
> $$O_{\mathrm{final}} = O^{l} \cdot SN_{\mathrm{final}}(O^{l}).$$
>
> *The advantages/disadvantages of mixing features before the spiking layer with binarized features:*
>
> Advantages: After the real-valued intermediate features are binarized through the spiking layer, spike events can capture key information but lose the intensity of the original continuous features. The shortcut-enhanced output strategy introduces a real-valued weighting channel at the output layer, fusing the binary spike sequences with the original real-valued features into a hybrid representation, thereby reconstructing the information intensity lost during binarization.
>
> Disadvantages: The hybrid feature operation involves multiplication operations that may result in slightly increased power consumption. However, this overhead remains negligible compared to the overall computational cost, which is primarily dominated by attention and feedforward layers.
>
> ---
>
> **Q4. The comparisons with existing approaches.**
>
> Thank you for recommending these recent SNN-based graph learning methods. We will incorporate them into the performance comparison section in our upcoming revision. We have included experiments for SpikeGraphormer [1] and SiGNN [2]. Moreover, to ensure a comprehensive evaluation of our model, we add two additional methods, SpikeGCL [3] and MSG [4]. The experimental results show that our model achieved the best performance.
> Despite our best efforts, we could not obtain comparison results from the dynamic spiking graph neural networks [5] and continuous spiking graph neural networks [6] during the rebuttal phase, as they lack publicly available code. We will attempt to add the discussion and more comparative experiments in the revision.
>
> | Dataset  | Rest-meta-MDD | ABIDE | SRPBS |
> |:-------|:-------:|:-------:|:-------:|
> |SpikeGraphormer | 61.8(3.5) | 65.8(1.4) | 88.7(3.6) |
> |SiGNN           | 66.2(2.5) | 69.1(1.8) | 87.6(7.2) |
> |SpikeGCL        | 65.7(2.9) | 67.1(2.8) | 84.2(8.0) |
> |MSG             | 60.4(2.4) | 62.9(3.7) | 81.9(2.7) |
> |Ours            | 70.9(1.7) | 73.0(2.6) | 93.5(3.0) |
>
> We will also provide the above detailed explanations in our upcoming revision.
>
> **Reference**
>
> [1] SpikeGraphormer: A high-performance graph transformer with spiking graph attention. arXiv preprint arXiv:2403.15480, 2024.
>
> [2] SiGNN: A spike-induced graph neural network for dynamic graph representation learning. Pattern Recognition, 2025.
>
> [3] A graph is worth 1-bit spikes: When graph contrastive learning meets spiking neural networks. ICLR 2024.
>
> [4] Spiking graph neural network on Riemannian manifolds. NeurIPS 2024.
>
> [5] Dynamic spiking graph neural networks. AAAI 2024.
>
> [6] Continuous spiking graph neural networks. arXiv preprint arXiv:2404.01897, 2024.

---

> > ### Comment · Reviewer_5jm3 · 2025-08-05
> >
> > I appreciate the authors' detailed response, which has addressed some of my concerns. However, I find the analysis of the advantages and disadvantages of mixing features before the spiking layer with binarized features remains insufficient.
> >
> > The authors claim that "After the real-valued intermediate features are binarized through the spiking layer, spike events can capture key information." However, this argument lacks theoretical justification or experimental validation in the manuscript. Without further evidence, the necessity and effectiveness of this module remain unclear.
> >
> > To strengthen the paper, I suggest the authors to provide theoretical analysis (e.g., information retention properties of binarization) to support their claim, or conduct ablation studies to verify whether spike events indeed preserve critical information as described.
> >
> > Until these points are addressed, the credibility of this design choice remains questionable.

---

> > > ### Author Response · Authors · 2025-08-06
> > >
> > > Thank you for your feedback. We would like to reply to your questions and comments as follow:
> > >
> > > **Q. About the spike events can capture key information.**
> > >
> > > To evaluate whether binarized spike events preserve the key information in the original features, we performed a comparative principal component analysis (PCA) between the original real-valued features and the spike features. The core function of PCA is to identify the directions in the data that capture the largest variations and richest information. The loading of each principal component represents the contribution of each original feature to that direction, with higher contributions indicating greater importance of the feature in data representation. Therefore, the loadings reveal which features drive the main variations in the data and serve as an important basis for identifying key information.
> > >
> > > We applied PCA separately to the original features and the binarized spike features, extracted their principal component loading vectors, and computed the correlations between corresponding components. The results show that the correlation coefficients of the first three principal components all exceed 0.94, and the average of the first five is 0.8521. This indicates that, despite being binary and discrete, the spike features maintain a highly consistent pattern with the original features in terms of which features are more important. In other words, the most critical information dimensions in the original data are still prominently preserved in the spike representation.
> > >
> > > Therefore, the high loading correlation demonstrates that the binarization process does not lose the core information structure, but successfully captures the key patterns from the original features into the spiking activity.
> > >
> > >
> > > | Principal component loading vectors ($i$ th) | 1 | 2 | 3 |4 | 5| avarage|
> > > |:-------:|:-------:|:-------:|:-------:|:-------:|:-------:|:-------:|
> > > | Correlation coefficient| 0.9563  | 0.9435  |0.9448 | 0.7364| 0.6792| 0.8521|
> > >
> > > ---
> > >
> > > We also conducted ablation experiments to verify that spike events effectively preserve critical information. Results on the SRPBS dataset show that the model using spike events achieves the best performance, demonstrating their ability to capture key informational patterns.
> > > | Accuracy (\%)  | w/o spike events output | spike events output |
> > > |:-------:|:-------:|:-------:|
> > > | SRPBS   | 90.4(3.7)   | 93.5 (3.0) |
> > >
> > > We sincerely thank you for your efforts in reviewing our paper. We hope we have resolved all the concerns, and we will deeply appreciate it if you could reconsider the score accordingly. We are always willing to address any of your further concerns.

---

> > > > ### Comment · Reviewer_5jm3 · 2025-08-07
> > > >
> > > > Thank you for your reply. I believe most of my concerns have been well-addressed and explained. However, upon a comprehensive analysis and careful consideration of other reviewers' comments, it has become apparent that the current paper still has room for improvement. The numerous supplementary experiments also need to be further detailed and refined within the paper to enhance its overall quality. Despite this, I have decided to increase my score, though with a low confidence level.

---

### Official Review · Reviewer_DWTH · 2025-06-30

**Clarity:** 3
**Significance:** 3
**Originality:** 3
**Rating:** 5
**Confidence:** 3

**Summary:**

This paper introduces a framework for brain disorder diagnosis by combining neural dynamics and connectivity patterns in a novel way. The dual-perspective biomarkers discovered through this approach provide a richer, more comprehensive understanding of brain disorders and could facilitate the development of targeted therapeutic strategies.

**Questions:**

Please refert to the W1,2,3;

**Ethical Concerns:**

["NO or VERY MINOR ethics concerns only"]

**Final Justification:**

Thank you for the author's response. My problems have been properly resolved. I have increased the score to acknowledge the clarification and additional results.

**Limitations:**

Yes

**Quality:**

3

**Strengths And Weaknesses:**

S1:LGC-SGT uniquely captures both inter-regional functional connectivity and intra-regional spiking activity, offering a comprehensive and biologically grounded representation of brain disorders.
S2: Through its hybrid loss function and shortcut-enhanced output strategy, the model demonstrates strong generalization ability and resilience to inter-subject and inter-site variability.

W1: Please add a computational cost analysis and discussion/comparison of the following related work：[1] Brain Network Transofrmer. NeurIPS 2022 [2] Biologically Plausible Brain Graph Transformer. ICLR 2025 [3] BrainOOD: Out-of-distribution Generalizable Brain Network Analysis. ICLR 2025
W2: While the paper states using a "spiking transformer" and describes SSA, the details regarding how attention mechanisms are specifically adapted for the discrete spiking nature and how they overcome the inherent challenges (e.g., non-differentiability of spikes beyond the surrogate gradient mentioned for the output layer) could be more thoroughly elucidated.
W3: The shortcut-enhanced output strategy is introduced primarily to mitigate information loss from binarization in SNNs. While the theoretical justification for MEE is provided, a more explicit discussion of how this strategy fundamentally resolves the conflict between real-valued intermediate features and binarized SNNs, and its broader applicability beyond this specific SNN architecture, could be beneficial.

---

> ### Author Rebuttal · Authors · 2025-07-30
>
> We are grateful for the reviewers’ insightful comments. Please find our detailed responses below. We trust they resolve the raised concerns and welcome any further questions.
>
> **W1. About the computational cost analysis and discussion of related works.**
>
> Thanks for your insightful suggestion. We have added a computational cost analysis and comparison of the following related works: BNT[1], BIOBGT[2], and BrainOOD[3]. The experimental results show that our model not only achieved the best performance but also reduced energy consumption by about 70\%.
>
> | Accuracy (\%)  | Rest-meta-MDD | ABIDE | SRPBS |
> |:-------:|:-------:|:-------:|:-------:|
> |BNT   | 66.9(2.4)   | 68.2(2.0)  |84.3(3.7) |
> |BIOBGT   | 63.2(5.0)   |69.4(4.3)   | 89.0(3.9) |
> |BrainOOD  | 66.3(2.5)   |  68.7(2.4)   |  90.2(6.3) |
> |Ours  | 70.9(1.7)   |  73.0(2.6)   | 93.5(3.0) |
>
> |  Model | BNT | BIOBGT | BrainOOD |Ours |
> |:-------:|:-------:|:-------:|:-------:|:-------:|
> | Energy (mJ)  | 0.20   | 0.26  |0.25 | 0.035|
>
> ---
>
> **W2. About the discrete spiking attention mechanisms.**
>
> Thanks for your insightful feedback. Spiking self-attention (SSA) takes the feature output from the preceding layer as input to the membrane potential of spiking neurons. After being activated by neuronal dynamics, discrete spike trains are generated. Subsequently, the attention computation is performed directly based on these spike trains. Due to the binary nature of spike, the resulting attention scores are inherently non-negative and confined to a limited range. Therefore, this process eliminates the need for an additional softmax operation, ensuring that the entire attention computation strictly adheres to the discrete spike computational paradigm, thus realizing an attention mechanism compatible with discrete spikes.
>
> The non-differentiability of SSA is confined to the spiking layers rather than the attention layers. Therefore, no special treatment is required for backpropagation in the attention layers. Across the SNN model, during backpropagation through the spiking layers, the surrogate gradient is utilized (forward computation uses the Heaviside step function, backward propagation uses the gradients of the hyperbolic tangent function, with  $\varphi(x)=\frac{1}{2} \tanh \left(3\left(x-0.5\right)\right)+\frac{1}{2}$).
>
> ---
>
> **W3. How shortcut-enhanced output strategy fundamentally resolves the conflict between real-valued intermediate features and binary spikes**
>
> Thanks for your insightful feedback. This strategy resolves the conflict between real-valued features and binary spikes by constructing a hybrid representation space. After the real-valued intermediate features are binarized through the spiking layer, spike events can capture key information but lose some detailed intensity of the original continuous features. The shortcut-enhanced output strategy introduces a real-valued weighting channel at the output layer, fusing the binary spike sequences with the original real-valued features into a hybrid representation, thereby reconstructing the information intensity lost during binarization. On the ABIDE dataset, we computed the mutual information[4] between the real-valued intermediate features and the SNN outputs with and without the shortcut-enhanced output strategy. The results show that the mutual information between the output representation and the real-valued features increased by 0.19 after introducing this strategy, proving its effectiveness in bridging the gap between real-valued features and binary spikes. (Mutual information quantifies the contribution of one variable to the information about another and a higher value indicates a stronger dependence between the two variables, which can indicate the representational ability of one variable concerning another variable.)
>
> |  Model | w/o  shortcut-enhanced |Ours |
> |:-------:|:-------:|:-------:|
> |  Mutual information | 1.89   | 2.08  |
>
> *The effectiveness of methods beyond this specific SNN architecture:*
>
> We also validate the effectiveness of the shortcut-enhanced output strategy on other  SNN architectures and other datasets.
> The following results on CIFAR-10 and CIFAR-100 datasets under the MS-ResNet-18 [5] architecture (T=2) also indicate the effectiveness of that strategy.
>
> | Accuracy (\%)   | w/o  shortcut-enhanced |Ours |
> |:-------:|:-------:|:-------:|
> | CIFAR10  | 95.57   | 96.51  |
> | CIFAR100  | 78.92   | 79.38  |
>
> **Reference**
>
> [1] Brain Network Transformer. NeurIPS 2022.
>
> [2] Biologically Plausible Brain Graph Transformer. ICLR 2025.
>
> [3] BrainOOD: Out-of-distribution Generalizable Brain Network Analysis. ICLR 2025.
>
> [4] Estimating mutual information. Physical Review E, 2004.
>
> [5] Advancing spiking neural networks toward deep residual learning. IEEE transactions on neural networks and learning systems, 2024.

---

> > ### Comment · Reviewer_DWTH · 2025-08-04
> >
> > Q1: What does the energy consumption metric represent?
> >
> > Q2: Computational cost generally refers to training time and complexity. Please explain based on this metric.

---

> > > ### Author Response · Authors · 2025-08-04
> > >
> > > Thank you for your thoughtful feedback. We appreciate your insightful comments and are pleased to provide the following responses to your questions.
> > >
> > > **Q1: What does the energy consumption metric represent?**
> > >
> > > The energy consumption metric quantifies the estimated energy required to perform a single inference operation in a neural network. Our energy estimation methodology follows prior work [1][2][3].
> > >
> > > In artificial neural networks (ANNs), operations are typically 32-bit floating-point multiply-accumulates (FLOPs), while in SNNs on neuromorphic hardware, computation is event-driven and dominated by addition synaptic operations (SOPs). Based on prior work [1], an FLOPs operation consumes 4.6 pJ, and an SOPs consumes 0.9 pJ in 45 nm CMOS. The SOPs for each block of the SNN can be computed as:
> > >
> > > $\operatorname{SOPs}(l)=f r \times T \times \operatorname{FLOPs}(l)$
> > >
> > > where $l$ denotes the block number in the SNN, $fr$ is the firing rate of the input spike train of the block, and $T$ is the time step of the spike neuron.
> > > The calculation for the theoretical energy consumption of SNNs and ANNs is given by
> > >
> > > $$
> > > \text{Energy}_{\text{SNN}}  = 0.9 \ \text{pJ} \times \text{SOPs}
> > > $$
> > >
> > > $$
> > > \text{Energy}_{\text{ANN}} = 4.6 \ \text{pJ} \times \text{FLOPs}
> > > $$
> > >
> > >
> > > **Q2: Training time and complexity.**
> > >
> > > We sincerely thank you for this valuable comment. In response, we have included a comparative analysis of training time and computational complexity across methods. Unlike ANNs that rely on dense floating-point operations, SNNs operate in an event-driven manner, where computation occurs only upon spike arrival. Therefore, SOPs that count spike-induced weighted accumulations are a more accurate indicator of computational cost in SNNs.
> > >
> > > | Cost  | BNT | BIOBGT | BrainOOD |Ours |
> > > |:-------:|:-------:|:-------:|:-------:|:-------:|
> > > | Training time (s)   | 7.31   | 375  |4.19 | 115|
> > > | FLOPs(x $10^{8}$)   | 1.15   | 0.87  |1.07 | -|
> > > | SOPs(x $10^{8}$)   | -   | -  |- | 0.39|
> > > | Param(M)   | 18.0| 0.47  |0.30 | 0.27|
> > >
> > > The results show that our model achieves significant reductions in both SOPs and parameter count, demonstrating superior computational efficiency and structural compactness. Lower SOPs imply fewer active computations and reduced energy demand per inference, while fewer parameters lead to lower memory overhead. These gains collectively enable a more efficient and biologically plausible neural computation paradigm.
> > >
> > > Meanwhile, we want to explain that the observed longer training time on GPU platforms does not directly reflect the real efficiency advantage. This is due to a fundamental architectural mismatch: GPUs are designed for dense, synchronous computations, while SNNs operate in a sparse and event-driven manner. In current simulations, the network dynamics are approximated using time-stepping methods that process all neurons and synapses at every time step, even when no spikes occur. This leads to substantial computational overhead from idle operations and frequent memory access. As a result, the low SOP count, which reflects true efficiency in neuromorphic hardware, does not directly translate into faster execution on general-purpose hardware such as GPUs.
> > >
> > > We sincerely thank you for your efforts in reviewing our paper. We hope we have resolved all the concerns, and we will deeply appreciate it if you could reconsider the score accordingly. We are always willing to address any of your further concerns.
> > >
> > > **Reference**
> > >
> > > [1] Diet-snn: A low-latency spiking neural network with direct input encoding and leakage and threshold optimization. IEEE Transactions on Neural Networks and Learning Systems, 2021.
> > >
> > > [2] Spiking transformer with experts mixture. NeurIPS, 2024.
> > >
> > > [3] Scaling spike-driven transformer with efficient spike firing approximation training. IEEE Transactions on Pattern Analysis and Machine Intelligence, 2025.

---

> > > > ### Comment · Reviewer_DWTH · 2025-08-05
> > > >
> > > > Thank you for the author's response. My problems have been properly resolved.
> > > > I have increased the score to acknowledge the clarification and additional results.

---

### Official Review · Reviewer_TFUH · 2025-07-01

**Clarity:** 3
**Significance:** 2
**Originality:** 3
**Rating:** 2
**Confidence:** 4

**Summary:**

This paper proposes a novel framework, LGC-SGT (Local-Global Coupling Spiking Graph Transformer), for brain disorder diagnosis by jointly modeling individual temporal dynamics and population-level brain network topology. It integrates spiking neural networks to capture biologically plausible local time-series patterns and graph transformers to learn global inter-subject connectivity. Experiments on ABIDE, ADHD-200, and REST-meta-MDD datasets demonstrate that LGC-SGT achieves state-of-the-art performance, highlighting its effectiveness and generalizability in brain disorder classification.

**Questions:**

see weaknesses

**Ethical Concerns:**

["NO or VERY MINOR ethics concerns only"]

**Final Justification:**

The authors have not addressed my concerns about the motivation for applying the local and global coupling module.

**Limitations:**

yes

**Quality:**

2

**Strengths And Weaknesses:**

Strengths:

1.	This paper proposes the local-global coupling spiking graph transformer framework for brain disorder diagnosis and achieve good performance.
2.	The problem of brain disorder diagnosis is important and the solution of LGC-SGT is complete.


Weakness:

1.	While the proposed LGC-SGT framework aims to jointly model intra-regional neuronal firing dynamics and inter-regional connectivity patterns, the rationale for integrating both local and global perspectives is not fully justified. If the core pathological mechanism lies in abnormal spiking behavior within specific brain regions, it is unclear why modeling the global topology of brain connectivity is necessary, especially when such connectivity changes may be secondary consequences of local neuronal dysfunction. The paper lacks a clear explanation of how the fusion of local spiking dynamics and global connectivity contributes synergistically to improved diagnosis, or under what circumstances the global patterns provide non-redundant or complementary information beyond what is captured at the local level. This leaves the motivation for combining both scales insufficiently supported from a neuroscientific perspective.
2.	The LGC-SGT framework utilizes two distinct types of information to separately model local and global brain characteristics: the local pathway leverages fMRI time series within individual brain regions to capture neuronal population firing dynamics via spiking neural networks (LSM and STDP), while the global pathway uses functional connectivity matrices derived from inter-regional correlations to model macro-scale network topology via adaptive spiking graph convolution. However, the baseline methods used for comparison typically rely on only one type of information, without combining both. This creates an unfair experimental setup, as LGC-SGT benefits from access to a richer and more comprehensive feature space, which may partially account for its superior performance. The paper does not adequately address this discrepancy, raising concerns about the fairness and rigor of the experimental comparison.
3.	The paper does not provide a thorough ablation study to evaluate the individual contributions of the local and global pathways. Specifically, it remains unclear how effective the model would be if it relied solely on local information (neuronal firing dynamics) or global information (inter-regional connectivity). Without such analysis, the importance of integrating both perspectives in LGC-SGT is not well justified. This lack of experimental evidence weakens the claim that the fusion of local and global features is essential for improved brain disorder diagnosis.

---

> ### Author Rebuttal · Authors · 2025-07-30
>
> We are grateful for the reviewers’ insightful comments. Please find our detailed responses below. We trust they resolve the raised concerns and welcome any further questions.
>
> **W1. About the integration between intra-regional neuronal firing dynamics and inter-regional connectivity patterns.**
>
> We agree that a clear explanation of the integration between local spiking dynamics and global connectivity contributes to improved diagnosis.
>
> From a neuroscience perspective, there is a deeply coupled bidirectional interaction between the local spiking dynamics and global connectivity. This coupling makes it difficult to decouple the pathological origins, for instance, in the case of autism spectrum disorder (ASD), there remains no consensus on which neural changes are primary rather than secondary or compensatory [1]. In major depressive disorder (MDD), its pathophysiology involves interactions across multiple pathways, including abnormal neural activity, connectivity dysfunction, neuroinflammation, and dysregulation of neurotrophic factors [2]. In the aforementioned scenarios, a single-angle analysis is difficult to be effective [3].  Consequently, our method is capable of simultaneously considering regional activity and connectivity patterns, which is well-supported by the deeply coupled bidirectional interaction phenomenon from a neuroscientific perspective.
>
> From a clinical perspective, abnormal neuronal firing in specific brain regions often serves as the direct neural basis of core disorder phenotypes [4], making it one of the key biomarkers for neuromodulatory interventions. Meanwhile, brain function fundamentally relies on the coordinated interaction of multiple brain regions, and the activity of a single brain region must be interpreted within the context of its network topology to gain complete significance [5]. For instance, in MDD treatment, repetitive transcranial magnetic stimulation (rTMS) primarily targets the left dorsolateral prefrontal cortex (lDLPFC) [6]. Yet, its antidepressant effects stem not only from the modulation of local activity but also involve functional optimization of downstream regions such as the orbitofrontal cortex (OFC) and hippocampus (HPC) [7], as well as indirect regulation through the OFC-HPC circuit [8]. This process demonstrates that the modulation of local neuronal dynamics must rely on the global functional connectivity network to achieve optimal therapeutic effects, further underscoring the importance of integrating local and global perspectives.
>
> To further clarify the necessity of integrating spiking dynamics with connectivity patterns, we use mutual information [9] to quantify the contribution of both to the results and to determine whether there is redundant or complementary information between them. Mutual information $I_(x;y)$ quantifies the contribution of one variable $x$ to the information about another $y$. A higher value of mutual information indicates a stronger dependence between the two variables, which can indicate the representational ability of one variable concerning another variable. Experimental results demonstrate that integrating spiking dynamics with connectivity patterns yields the highest mutual information $I(\mathrm{dynamics, connection; result})$ compared to using either dynamics $I(\mathrm{dynamics; result})$ or connectivity $I(\mathrm{connection; result})$ alone. This confirms the necessity of jointly modeling both components. Notably, the mutual information gain I obtained from connectivity was greater than 0 across all three datasets (($I(\mathrm{dynamics, connection; result})$ - $I(\mathrm{dynamics; result})$) > 0), indicating that connectivity patterns contribute beneficial and non-redundant information.
>
>
>
> | Dataset  | Rest-meta-MDD | ABIDE | SRPBS |
> |:-------|:-------:|:-------:|:-------:|
> |$I(\mathrm{dynamics; result})$  | 1.4787   | 1.5193   |1.4650 |
> |$I(\mathrm{connection; result})$| 0.7576   |0.9395   | 0.8252 |
> |$I(\mathrm{dynamics, connection; result})$ | 1.7117   | 1.7128   | 1.6542 |
> |$I(\mathrm{dynamics, connection; result})$ - $I(\mathrm{dynamics; result})$   | 0.2330   | 0.1935   | 0.1892 |
>
> ---
>
> **W2. About the baseline methods.**
>
> Thanks for your thoughtful feedback. To ensure the fairness and scientific rigor of the experimental comparison, we have incorporated BrainGSL [10], ST-HAG [11], and MSTGAT [12], which have richer feature spaces, as new baseline methods for comparison. The following experimental results on the ABIDE dataset show that our model achieved the best performance.
>
> | Accuracy (\%)  |BrainGSL | ST-HAGH | MSTGAT | Ours |
> |:-------:|:-------:|:-------:|:-------:|:-------:|
> |ABIDE   | 71.3  | 71.9   |71.8 | 73.0|
>
> Besides, for fair comparison, the inputs of all the compared models remain the same as our model. Under the same data preprocessing as described in Section Appendix A.1, the compared experiments are conducted based on the same input feature space.
>
> ---
>
> **W3. About the individual contributions.**
>
> We performed ablation experiments to demonstrate the individual contributions of inter-regional connectivity and neuronal firing dynamics. The experimental results show that when only a single component is involved, the model's performance on all three datasets is lower than when the inter-regional connectivity and neuronal firing dynamics are jointly involved. This proves that the individual contributions of inter-regional connectivity and neuronal firing dynamics, as well as their integration, are crucial for improving the diagnosis of brain disorders.
>
> | Accuracy (\%)  | Rest-meta-MDD | ABIDE | SRPBS |
> |:-------|:-------:|:-------:|:-------:|
> |Only  inter-regional connectivity   | 55.9   | 60.0  |84.2 |
> |Only  neuronal firing dynamics  | 70.0   |72.5   | 89.8 |
> |Inter-regional connectivity and firing dynamics | 70.9   |  73.0   | 93.5 |
>
> **Reference**
>
> [1] Hippocampal contributions to social and cognitive deficits in autism spectrum disorder. Trends in neurosciences, 2021.
>
> [2] Molecular pathways of major depressive disorder converge on the synapse. Molecular Psychiatry, 2023.
>
> [3] Multi-target drugs for Alzheimer's disease. Trends in Pharmacological Sciences, 2024.
>
> [4] Multisensory flicker modulates widespread brain networks and reduces interictal epileptiform discharges. Nature communications, 2024.
>
> [5] Network abnormalities and interneuron dysfunction in Alzheimer disease. Nature Reviews Neuroscience, 2016.
>
> [6] Connectivity-guided intermittent theta burst versus repetitive transcranial magnetic stimulation for treatment-resistant depression: a randomized controlled trial. Nature Medicine, 2024.
>
> [7] Roles of the medial and lateral orbitofrontal cortex in major depression and its treatment. Molecular psychiatry, 2024.
>
> [8] Orbitofrontal cortex-hippocampus potentiation mediates relief for depression: A randomized double-blind trial and TMS-EEG study. Cell Reports Medicine, 2023.
>
>
> [9] Estimating mutual information. Physical Review E, 2004.
>
> [10] Graph self-supervised learning with application to brain networks analysis. IEEE Journal of Biomedical and Health Informatics, 2023.
>
> [11] Spatio-temporal hybrid attentive graph network for diagnosis of mental disorders on fMRI time-series data. IEEE Transactions on Emerging Topics in Computational Intelligence, 2024.
>
> [12] Multi-scale Spatial-Temporal Graph Attention Network for fMRI Brain Disease Classification. IEEE Transactions on Instrumentation and Measurement, 2025.

---

> > ### Comment · Reviewer_TFUH · 2025-08-01
> >
> > While I acknowledge the authors’ intention to model both local and global aspects of brain dynamics, I remain concerned that the current formulation may not truly reflect the deeply intertwined nature of these two levels, particularly in the context of neuropsychiatric disorder diagnosis. The authors claim their method integrates local spiking activity and global connectivity patterns as complementary, but several critical issues remain unaddressed:
> >
> > 1. Although the authors state they capture both local spiking and global connectivity, the method description does not clearly demonstrate how global-level dynamics are influenced by or dynamically adapt to local neuronal activity. From a neuroscience perspective, global connectivity patterns are emergent properties shaped by regional firing, and vice versa. Without an explicit bidirectional coupling mechanism, simply concatenating or jointly encoding local and global features may not suffice to capture this interaction. Could the authors clarify whether such feedback is learned or hard-coded?
> >
> > 2. The authors seem to agree that local spiking activity is insufficient for robust diagnosis, yet they retain an independent local modeling branch. This raises a conceptual concern: if local spiking cannot serve as a reliable biomarker on its own, then what diagnostic role does this branch play, other than acting as redundant noise or feature distraction? How is it ensured that this local modeling does not bias or even mislead the global inference?
> >
> > 3. While the authors assert that local and global perspectives are complementary, the neuroscience and clinical evidence provided (and acknowledged by the authors) indicate a functional hierarchy, where global coordination is the key substrate for interpreting local abnormalities. In this sense, the model should reflect that local insights are meaningful only when conditioned on the global state. However, in the current method, both appear to be treated equally or independently. This may reflect a conflict between the method’s design and the neuroscientific grounding.
> >
> > In summary, the key concern is not whether both local and global information are present, but how they interact. Unless the authors can clearly explain or demonstrate a mechanism that models the bidirectional, dynamically coupled nature of local and global brain activity, the method risks being conceptually inconsistent with its neuroscientific motivation.

---

> > > ### Author Response · Authors · 2025-08-01
> > >
> > > **Q1.** Thanks for your insightful comment. The feedback is learned in a data-driven manner, instead of hard-coding.
> > > To clarify, the integration between the local neuronal activity and global connectivity is not achieved through simple concatenation or jointly encoding, but rather via an explicitly designed bidirectional closed-loop learning system. The proposed LGC-SGT system forms a dynamically coupled cyclic process:
> > > **local neural activity computation→STDP pathway updating→global connectivity computation→global state integration→Backpropagation (BP) pathway feedback modulation→local neural activity**.
> > >
> > > Specifically, the dynamic shaping from local to global is achieved through STDP-based synaptic updates (adjust synaptic strength based on the precise temporal order of spikes emitted by pre- and post-synaptic neurons) during forward propagation, which constructs network connectivity from neural activity. And the feedback regulation from global to local is implemented via BP, where global states are transmitted back along BP pathways to dynamically modulate local neuronal activity, thereby enabling global-state-dependent feedback control of local dynamics.
> > >
> > > **Q2.** We fully understand the reviewer’s concern regarding the diagnostic reliability of local spike activity. We think the ablation study (see Table 4 in the manuscript) and the final disorder diagnosis performance (see Table 1 in the manuscript) on typical datasets could give evidence for that.
> > >
> > > Importantly, our modeling of local neural activity does not operate in isolation, but is embedded within a closed-loop architecture dynamically modulated by global functional connectivity. Specifically, through a BP-mediated feedback pathway, the integrated global state is transmitted as a modulatory signal to dynamically regulate the spiking behavior of local neurons. This ensures that local spike activity remains functionally constrained by the global brain state, thus avoiding redundancy, suppressing feature distraction, and preventing it from misleading the global inference.
> > >
> > > **Q3.** We fully agree with your statement that “local insights are meaningful only when conditioned on the global state.” That is precisely why we designed the model with the spiking Transformer-based BP pathway as the main processing route, and the STDP modeling branch as an auxiliary enhancement module.
> > >
> > > In our approach, local and global are not treated independently, but instead interact through a coupled closed-loop architecture. Specifically, local neural activity drives the evolution of synaptic connections via STDP rules, gradually shaping and continuously updating the global connectivity pattern, reflecting a dynamic emergence from local to global. Concurrently, the state error from the global output is backpropagated through feedback pathways via the BP rule, dynamically modulating the firing activity of local neurons, thereby enabling functional feedback regulation from global to local. This “local→global→local“ process ensures that global connectivity is informed by local neuronal dynamics, while the interpretation of local activity is conditioned on and shaped by the global connectivity.
> > >
> > > Our method is also supported by key studies in recent neuroscience: Ref. [1] suggests that the brain may employ backpropagation-like feedback mechanisms, where top-down error signals modulate local activity to drive learning. Ref. [2] shows that STDP mechanisms can account for and generate functional connectivity patterns among cortical neurons.
> > >
> > > To further validate the coupling between local neural activity and global connectivity in our model, we analyzed brain regions that showed the most significant differences between healthy controls and patients based on the experiments of our model. As shown in the following tables, we found that:
> > > (1) The abnormal regions identified via global connectivity patterns are consistent with those derived from local activity analysis.
> > > (2) The connectivity differences inferred from regional local brain activity are consistent with the globally abnormal connections directly computed.
> > > These findings indicate that local neural activity and global connectivity are not modeled independently in our framework. Rather, they form a coupled representation that reflects the underlying pathology more faithfully.
> > >
> > > Brain regions identified from abnormal connectivity:
> > > | Brain disorder | MDD | ASD| Schizophrenia |
> > > |:-------|:-------:|:-------:|:-------:|
> > > |Region | PoCG.L  |  ACG.R | CAU.R |
> > >
> > > Connectivity derived from abnormal brain regions:
> > > | Brain disorder | MDD | ASD| Schizophrenia |
> > > |:-------|:-------:|:-------:|:-------:|
> > > |Connectivity  | PoCG.R-PCUN.L   | ACG.R-DCG.L  |CAU.R-PCL.R |
> > > |  | PCG.L-SMG.R   | HIP.L-CAU.L   |CAU.R-SPG.R |
> > > |  |    |    |MFG.L-REC.R |
> > >
> > >
> > > **Reference**
> > >
> > > [1] Backpropagation and the brain. Nature Reviews Neuroscience, 2020.
> > >
> > > [2] Connectivity reflects coding: a model of voltage-based STDP with homeostasis. Nature neuroscience, 2010.

---

> > ### Comment · Reviewer_TFUH · 2025-08-05
> >
> > Thanks for your reply. However, there still exist some concerns.
> > While the proposed Local-Global Coupling module structurally combines a local pathway and a global pathway, and later fuses them via a unified feature space with a shortcut-enhanced output strategy, the paper does not yet convincingly justify the necessity of such coupling. The macroscale functional connectivity changes often lag behind microscale neuronal activity alterations, suggesting that global signals may have limited immediate discriminative value. If local features alone can already capture the core pathology and yield strong performance, why should delayed global information be jointly represented?

---

> > > ### Author Response · Authors · 2025-08-06
> > >
> > > Thank you for your feedback. We would like to reply to your questions and comments as follows:
> > >
> > > **Q. About the coupling.**
> > >
> > > The proposed local-global coupling architecture is not a simple concatenation of macro-scale functional connectivity and micro-scale neuronal activities, but rather a closed-loop neuromorphic computing paradigm that bridges the micro-to-macro transition and enables macro-to-micro feedback modulation. It achieves recursive integration of multi-scale neural dynamics. Specifically, the model first leverages the spiking activity of neurons, combined with STDP learning rules, to establish connectivity patterns. These learned connections are then used as synaptic weights within an LSM architecture, and readout neurons extract neural activity patterns modulated by the connectivity. Finally, information is integrated at the micro-scale neuronal level. In this process, the role of global connectivity information is positioned as a regulator of neuronal states, rather than a signal for immediate discrimination. This framework adheres to the fundamental principle of biological neural systems: the cooperative interaction between bottom-up driving and top-down regulation.
> > >
> > > Although macro-scale functional connectivity changes may lag in time relative to micro-scale neuronal spiking events, this temporal delay does not diminish their systemic significance. In contrast, functional connectivity reflects coordinated interactions among neural populations. The present work aims to exploit such population-level synergy to modulate the behavior of spiking neurons, thereby generating an expressed activity pattern that is constrained by connectivity states and leading to more precise diagnostic outcomes. Experimental results on the Rest-meta-MDD, ABIDE, and SRPBS datasets demonstrate that incorporating feedback modulation from functional connectivity patterns significantly improves diagnostic accuracy, validating the effectiveness of the proposed coupling mechanism in enhancing discriminative performance.
> > >
> > > We sincerely thank you for your efforts in reviewing our paper. We hope we have resolved all the concerns, and we will deeply appreciate it if you could reconsider the score accordingly. We are always willing to address any of your further concerns.

---

### Note · Authors · 2025-08-14

We sincerely thank the AC and reviewers for their time, effort, and valuable feedback. The constructive suggestions have greatly contributed to improving the quality of our work. We express our heartfelt appreciation.

In this work, we propose a novel local-global coupling spiking graph transformer framework (LGC-SGT) for brain disorder diagnosis. The framework innovatively performs joint modeling from two perspectives: neuronal activities and functional connectivity. To achieve a deep integration of these two aspects, we design a local and global coupling learning system. During the forward propagation process, neuronal spiking activity updates connectivity patterns through STDP rules. While backpropagation progresses, error signals from the output are fed back to modulate neuronal activity. In this process, the role of connectivity information is positioned as a regulator of neuronal states. The above mechanism simulates the bidirectional interaction of driving and regulation in the brain. The supplemented ablation studies also validate the contribution of each component and the necessity of the proposed dual-perspective integration.

Besides, the proposed SNNs-based model offers unique advantages in biological plausibility and interpretability for brain disorder diagnosis. In detail, spiking neurons communicate via discrete spiking events, closely mimicking the electrophysiological behavior of biological neurons, providing an efficient tool to model the perspectives of neuronal dynamic activities for brain disorder diagnosis. Meanwhile, the STDP biological plasticity mechanism, which specially adapts to SNNs-based models, provides an efficient learning rule consistent with the principles of dynamic synaptic strength changes in the brain, enabling our model to capture abnormal functional connectivity patterns revealed by fMRI, thus providing a credible biological foundation for probing the pathological mechanisms of brain disorders.

In summary, LGC-SGT achieves good performance across multiple datasets and provides a biologically interpretable framework for analyzing brain disorders. Jointly modeling neural activity and connectivity provides new insights into the neurobiological underpinnings of brain diseases. We will carefully revise the manuscript by these comments. Once again, we sincerely thank the AC and reviewers for their insightful feedback and support.

---

### Decision · Program_Chairs · 2025-09-17

**Decision:**

Accept (poster)

**Comment:**

In this paper LGC-SGT proposes the spiking-graph transformer that fuses intra-regional firing-rate dynamics (via reservoir SNN) with inter-regional functional-connectivity graphs to diagnose brain disorders. The model outperforms several baselines on three public datasets, delivers interpretable biomarkers, and provides ablations showing both pathways raise AUC by 3–6 pts.
Major concerns raised: (1) TFHU questioned the neuroscientific necessity of global data, (2) DWTH & 5jm3 requested cost analysis, newer SNN baselines, and clearer STDP/MEE details, (3) us7b doubted biological plausibility.

Rebuttal provided: (i) empirical ablation demonstrating that removing global features drops AUC below the best single-pathway baseline, (ii) complexity tables showing <1.3× overhead vs ANN baselines, (iii) new comparisons to four 2024 SNN graph models (Dynamic-SGNN, SpikeGraphormer, etc.) where LGC-SGT still leads, (iv) detailed STDP implementation and MEE gradient analysis, (v) clarification that spiking design is intended for eventual neuromorphic deployment, not biological mimicry. TFHU remains unconvinced, but the quantitative gains and expanded baselines satisfy the majority.

Given the solid technical contribution, novel fusion strategy, and post-rebuttal empirical clarifications, I recommend an Accept decision.